# A multi-layered and dynamic apical extracellular matrix shapes the vulva lumen in *Caenorhabditis elegans*

**Jennifer D Cohen**[1], **Alessandro P Sparacio**[1], **Alexandra C Belfi**[1],
**Rachel Forman-Rubinsky**[1], **David H Hall**[2], **Hannah Maul-Newby**[3], **Alison R Frand**[3],
**Meera V Sundaram**[1]*

[1]Department of Genetics, University of Pennsylvania Perelman School of Medicine, Philadelphia, United States; [2]Department of Neuroscience, Albert Einstein College of Medicine, Bronx, United States; [3]Department of Biological Chemistry, David Geffen School of Medicine, University of California, Los Angeles, Los Angeles, United States

**Abstract** Biological tubes must develop and maintain their proper diameter to transport materials efficiently. These tubes are molded and protected in part by apical extracellular matrices (aECMs) that line their lumens. Despite their importance, aECMs are difficult to image in vivo and therefore poorly understood. The *Caenorhabditis elegans* vulva has been a paradigm for understanding many aspects of organogenesis. Here we describe the vulva luminal matrix, which contains chondroitin proteoglycans, Zona Pellucida (ZP) domain proteins, and other glycoproteins and lipid transporters related to those in mammals. Confocal and transmission electron microscopy revealed, with unprecedented detail, a complex and dynamic aECM. Different matrix factors assemble on the apical surfaces of each vulva cell type, with clear distinctions seen between Ras-dependent (1°) and Notch-dependent (2°) cell types. Genetic perturbations suggest that chondroitin and other aECM factors together generate a structured scaffold that both expands and constricts lumen shape.

*For correspondence:
sundaram@pennmedicine.upenn.edu

## Introduction

During tubulogenesis, lumen formation and expansion generally occur in the context of fluid influx and/or apical extracellular matrix (aECM) secretion (reviewed by *Luschnig and Uv, 2014*; *Navis and Nelson, 2016*). Tubular epithelia drive water into the lumen by establishing ionic and osmotic gradients using various ion pumps and channels; the resulting hydrostatic pressure can stimulate lumen enlargement (*Bagnat et al., 2007*; *Dong et al., 2011*; *Khan et al., 2013*; *Kolotuev et al., 2013*; *Navis et al., 2013*). But ions and water are not the only molecules being secreted into nascent lumens; proteoglycans, lipids, mucins, zona pellucida (ZP) domain proteins, and/or other matrix factors are also present and can contribute to lumen shaping (*Devine et al., 2005*; *Gill et al., 2016*; *Hwang et al., 2003b*; *Jaźwińska et al., 2003*; *Lane et al., 1993*; *Rosa et al., 2018*; *Strilić et al., 2009*; *Tonning et al., 2005*). These aECM factors may act like sponges to bind and organize water molecules and generate outward pushing forces (*Lane et al., 1993*; *Syed et al., 2012*), or they may assemble into fibrils or other specialized structures to exert more localized pushing or pulling forces on tube membranes (*Andrew and Ewald, 2010*; *Linde-Medina and Marcucio, 2018*; *Luschnig and Uv, 2014*; *Plaza et al., 2010*). aECMs may also bind and present or sequester various signaling molecules that impact cell identity or behavior (*Judge and Dietz, 2005*; *Perrimon and Bernfield, 2000*). aECMs of varying types are present in all tubular epithelia; examples in mammals include the vascular glycocalyx, lung surfactant, and the mucin-rich linings of the gastrointestinal tract and upper

airway (*Bernhard, 2016*; *Johansson et al., 2013*; *Webster and Tarran, 2018*). However, such aECMs generally appear translucent by light microscopy and are easily destroyed by standard chemical fixation protocols, and thus the organizational structures and lumen-shaping mechanisms of most luminal matrices remain poorly understood.

Vulva development in the nematode *Caenorhabditis elegans* has been a paradigm for understanding many aspects of cell fate specification and organogenesis (*Schindler and Sherwood, 2013*; *Schmid and Hajnal, 2015*). The vulva tube consists of twenty-two cells of seven different cell types, organized as seven stacked toroids (vulA, vulB1, vulB2, vulC, vulD, vulE, and vulF) (*Sharma-Kishore et al., 1999*; *Figure 1*). In adult hermaphrodites, the vulva connects to the uterus and serves as a passageway to allow sperm entry and the release of fertilized eggs. In the 40+ years since vulva cell lineages were first described (*Sulston and Horvitz, 1977*), much has been learned about how different vulva cell fates are specified and how they arrange to form the tube structure. It is known that the glycosoaminoglycan (GAG) chondroitin promotes initial expansion of the vulva lumen during morphogenesis (*Hwang et al., 2003a*), the lumen changes shape and eventually narrows, and then later, in the adult, collagenous cuticle lines the functional vulva tube (*Page, 2007*; *Sulston and Horvitz, 1977*). However, the specific contents, organization, and morphogenetic roles of the luminal matrix within the developing vulva have remained, for the most part, uncharacterized.

Here, we show that a spatially and temporally dynamic aECM assembles and disassembles within the vulva lumen during morphogenesis. This transient aECM shares components with the glycocalyx-like sheath or pre-cuticle matrix that coats other apical surfaces in *C. elegans* prior to each round of cuticle secretion (*Forman-Rubinsky et al., 2017*; *Gill et al., 2016*; *Katz et al., 2018*; *Kelley et al., 2015*; *Labouesse, 2012*; *Lažetić and Fay, 2017*; *Mancuso et al., 2012*; *Priess and Hirsh, 1986*; *Vuong-Brender et al., 2017*). It contains both fibrillar and granular components, and also extracellular vesicles, as observed at the ultrastructural level. Different combinations of matrix factors assemble on the apical surfaces of each of the seven different vulva cell types, with particularly clear distinctions seen between Ras-dependent (1°) and Notch-dependent (2°) cell types. Genetic perturbation experiments suggest that chondroitin and other aECM factors together generate a structured scaffold that has both lumen-expanding and lumen-constraining roles.

## Results

### Background: vulva tube formation

Specification and generation of the seven vulva cell types occur during the L2 and L3 larval stages, while toroid formation and other aspects of tube morphogenesis occur during the L4 stage (*Figures 1* and *2*). The 22 cells of the vulva are derived from three of six total possible vulva precursor cells (VPCs), named P3.p-P8.p. Signaling by the epidermal growth factor receptor (EGFR)-Ras-ERK and Notch pathways specifies one central 1° and two flanking 2° VPC fates (*Figure 1B*; *Schmid and Hajnal, 2015*; *Sternberg and Horvitz, 1989*). First, an EGF-like signal from the gonadal anchor cell (AC) induces P6.p to adopt the 1° VPC fate, and that cell then expresses Delta/Serrate/LAG-2 (DSL)-like ligands to induce its neighbors, P5.p and P7.p, to adopt the 2° cell fate. The 1° VPC (P6.p) divides to generate eight descendants: four vulF and four vulE cells. The 2° VPCs (P5.p and P7.p) divide to generate seven descendants each: one vulD, two vulC, one vulB2, one vulB1 and 2 vulA cells. Primary descendants produce an unknown cue that promotes basement membrane invasion by the gonadal AC, which is the first step in forming a vulva-uterine connection (*Ihara et al., 2011*; *Matus et al., 2014*; *Sherwood and Sternberg, 2003*). As they divide, 1° descendants detach from the underlying epidermal cuticle and move dorsally, and also detach from the overlying basement membrane to allow further AC penetration (*Ihara et al., 2011*; *Matus et al., 2014*; *McClatchey et al., 2016*). Upon completion of vulva cell divisions, 2° descendants migrate inward in a Rac and Rho-dependent manner, and push the more central 2° descendants and the 1° descendants further dorsally to generate the vulva invagination seen at early L4 (*Farooqui et al., 2012*; *Kishore and Sundaram, 2002*; *Sharma-Kishore et al., 1999*; *Shemer et al., 2000*). As cells of the same type meet, they eventually fuse to form the seven vulva toroids (*Sharma-Kishore et al., 1999*).

Ten different stages of L4 vulva morphogenesis (L4.0-L4.9) have been distinguished based on changing lumen morphology, as observed by differential interference contrast (DIC) microscopy (*Mok et al., 2015*). To visualize cell shapes that correspond to each of these stages, we used the

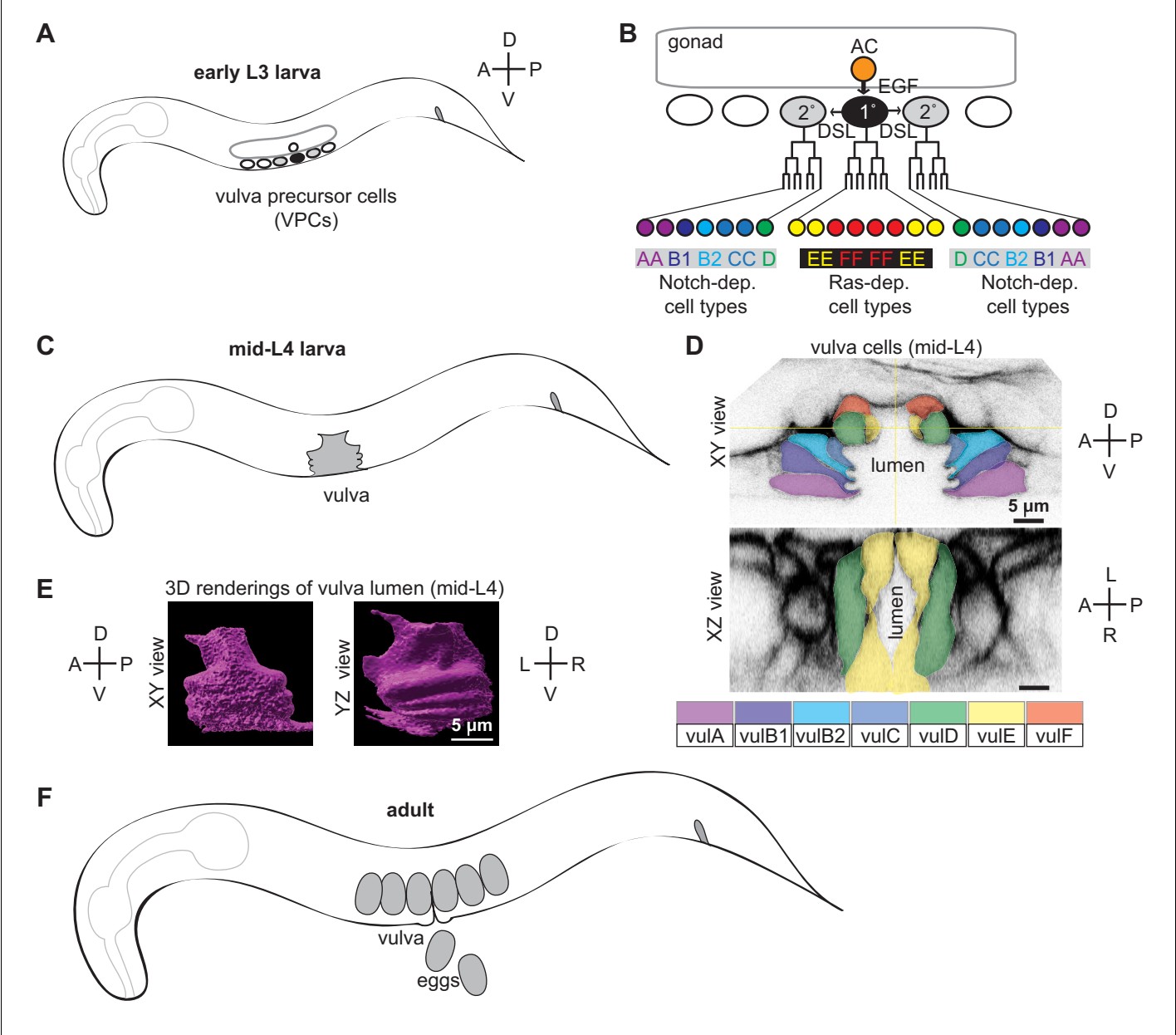

**Figure 1.** Introduction to vulva development. (**A**) Cartoon of early L3 larva, indicating the six vulva precursor cells (VPCs) beneath the somatic gonad. (**B**) Vulva lineages and cell types. An EGF-like signal from the gonadal anchor cell (AC) induces the primary (1˚) cell fate in the nearest VPC (black), which then expresses DSL ligands to induce the secondary (2˚) cell fate in the adjacent VPCs (gray). The 1˚ and 2˚ VPCs divide to generate a total of 22 descendants of 7 different cell types. (**C**) Cartoon of mid L4 larva, showing the vulva lumen. (**D**) L4.4 stage vulva cells visualized with the membrane marker MIG-2::GFP (*muIs28*). The 22 vulval cells are organized into 7 stacked rings (*Sharma-Kishore et al., 1999*). In the standard lateral or sagittal view, anterior is to the left and ventral is down. An orthogonal XZ view shows the oblong shape of the lumen. (**E**) 3D rendering of the L4.4 vulva lumen generated with Imaris software (BitPlane, Zurich Switzerland), based on imaging of the matrix factor FBN-1 (see *Figure 3*). The YZ view at right is comparable to the transverse views of the vulva seen by TEM in *Figures 4B* and *9C* (but note that regions deeper in the Z plane are poorly resolved here). (**F**) The adult vulva is a slit-like and cuticle-lined passageway through which eggs are laid.

RhoG marker MIG-2::GFP (*Honigberg and Kenyon, 2000*) to label all vulva cell membranes (*Figures 1D* and *2*). At L4.0, the vulva invagination is very narrow, but it enlarges to approximately 10 microns in diameter by the L4.3 stage. Between L4.3 and L4.4, the uterine lumen and the dorsal-most part of the vulva lumen both expand (*Ihara et al., 2011*; *Matus et al., 2014*), and the gonadal AC fuses with the uterine seam (utse), leaving just a thin part of its membrane as a hymen separating

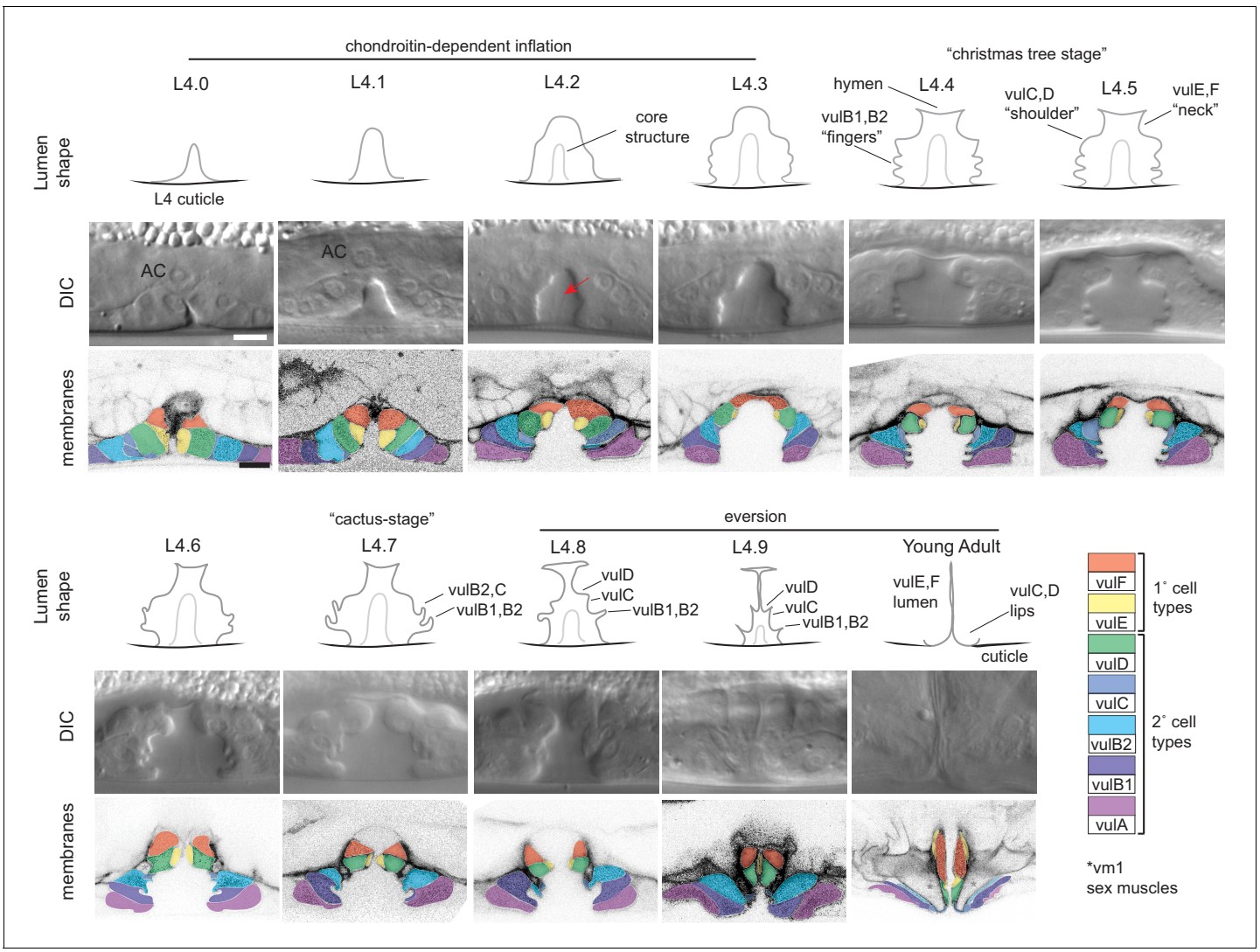

**Figure 2.** Cell and lumen shape changes during vulva morphogenesis Sagittal views of the central vulva lumen. Top rows show cartoons of lumen shape for each L4 sub-stage, as defined by *Mok et al., 2015*. Middle rows show corresponding DIC images. Bottom rows show confocal slices of vulva cell membranes marked by MIG-2::GFP (*muIs28*); cells are colored according to the key at right. Confocal stacks were collected for at least three animals per stage after L4.3. Luminal core structure is faintly visible beginning at L4.2 (red arrow). At mid-L4 ('Christmas-tree stage'; *Seydoux et al., 1993*), the vulF and vulE cells together define the vulva 'neck', the vulD and vulC cells define the vulva 'shoulder', the vulB1 and vulB2 cells define the vulva 'fingers', and the vulA cells make the connection between the vulva cells and the surrounding epidermis. *vm1 sex muscles, which attach to the mature vulva between the vulC and vulD toroids (*Sharma-Kishore et al., 1999*). Scale bars, five microns.

the two lumens (*Sapir et al., 2007*; *Sharma-Kishore et al., 1999*). The vulB1 and vulB2 cells also develop increasingly concave apical surfaces, creating a 'Christmas-tree-like' lumen appearance at L4.4-L4.5, and a more 'cactus-like' appearance by L4.7. In the final morphogenetic stages, collectively termed eversion, further cell shape changes and rearrangements occur to narrow the lumen and generate the closed lips of the final adult tube structure (*Seydoux et al., 1993*) (see below). Following eversion, the vulva lumen remains in a closed conformation unless opened by contractions of the sex muscles, which attach to multiple vulva toroids (*Sharma-Kishore et al., 1999*). During L4 and adult stages, each vulva cell type expresses different combinations of known transcription factors, membrane fusogens, or other molecular markers (*Inoue et al., 2004*; *Inoue et al., 2002*; *Inoue et al., 2005*; *Mok et al., 2015*; *Shemer et al., 2000*; *Sternberg and Horvitz, 1989*), but the biological distinctions among the seven cell types are not well understood.

A chondroitin proteoglycan (CPG)-rich luminal matrix is thought to form at the earliest stages of vulva tube morphogenesis, and to swell with water to exert a uniform pushing force for the lumen

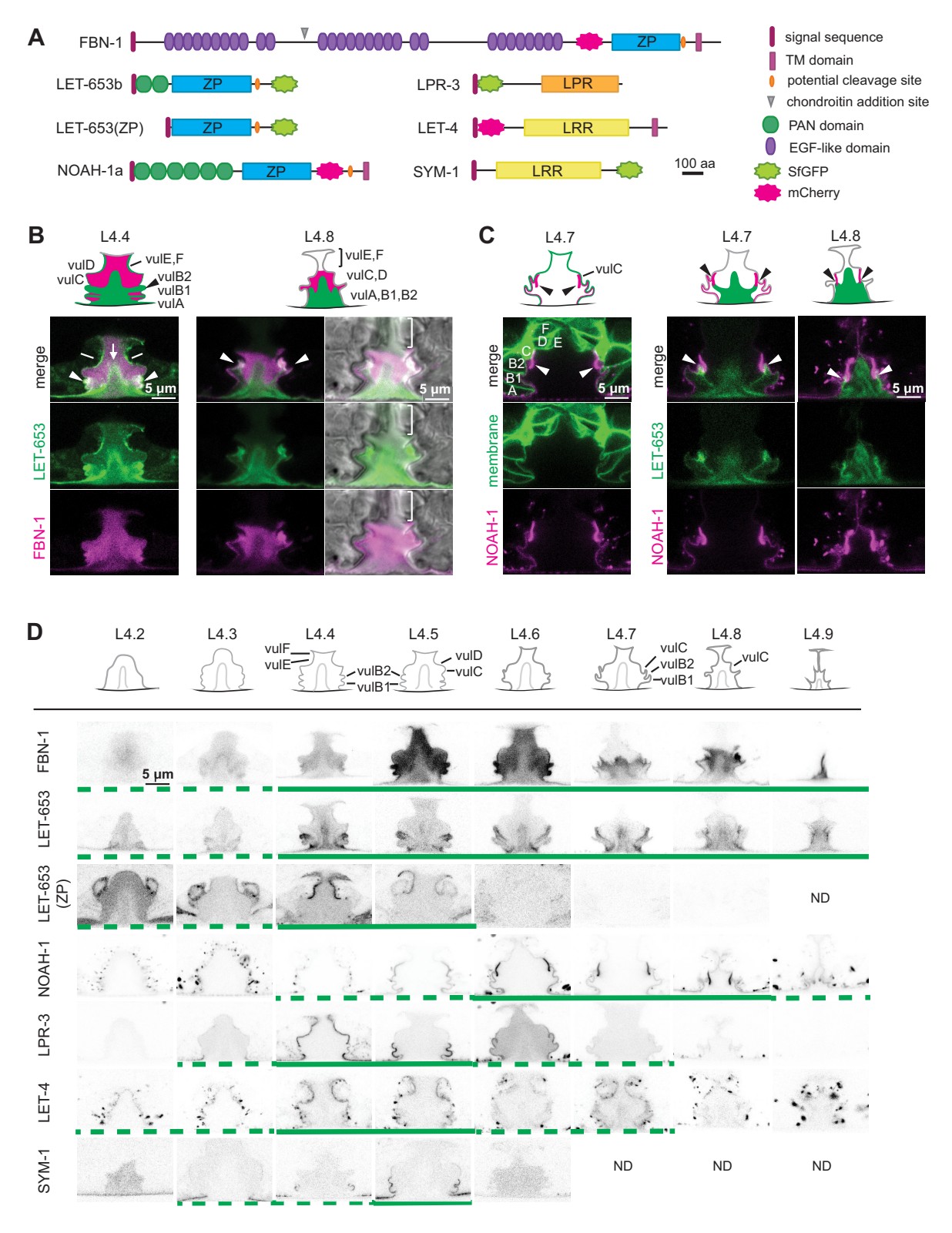

**Figure 3.** A dynamic aECM fills the vulva lumen during morphogenesis. (A) aECM protein schematics. Genbank accession: FBN-1a (AFN70749.1), LET-653b (CAH60755.1), NOAH-1a (CCD66686.2), LPR-3 (CAA92030.1), LET-4 (AEZ55699.1), SYM-1 (CAB43345). Full-length FBN-1::mCherry and some LET-653 fusions were expressed from transgenes; all others were expressed from the endogenous loci tagged by CRISPR-Cas9 genome editing (see Materials and methods). (B) LET-653::SfGFP (*csIs64*) and FBN-1::mCherry (*aaaIs12*) show complementary luminal patterns. Medial confocal slices. Arrow,

*Figure 3 continued on next page*

**Figure 3 continued**

luminal core. Arrowheads, sites of ventro-lateral fibril attachment to vulva cells. Lines, membrane-proximal matrix over 1° cells vulE and vulF. Bracket indicates loss of FBN-1 from the lumen over 1° cells during vulva eversion (n = 3/5 L4.7, 4/4 L4.8). (C) NOAH-1::mCherry (*mc68*) labels matrix spikes that connect to LET-653-marked luminal fibrils during vulva eversion. Left column shows overlay with cell membrane marker MIG-2::GFP (*muIs28*) (n = 9 L4.7-L4.9). Right columns show overlay with LET-653::SfGFP (*cs262*) (n = 12 L4.7-L4.9). (D) Timeline of vulva morphogenesis showing dynamic matrix patterns. Each image is a single confocal slice, inverted for clarity. For each fusion, images were collected for at least three animals per stage per strain; most fusions were imaged in multiple different strains to directly compare the different patterns (as in panels B and C). Solid green underlines indicate stages with consistent and peak localization; dashed green underlines indicate stages with more variable or weak localization. Fusions shown are FBN-1::mCherry (*aaaIs12*), LET-653::SfGFP (*cs262*), LET-653(ZP)::SfGFP (*csIs66*), SfGFP::LPR-3 (*cs250*), NOAH-1::mCherry (*mc68*), mCherry::LET-4 (*cs265*), and SYM-1::GFP (*mc85*).

expansion (*Gupta et al., 2012*; *Hwang et al., 2003a*). Chondroitin antibodies stain the mid-L4 vulva lumen, though in a disorganized fashion that likely reflects matrix destruction by chemical fixatives (*Bender et al., 2007*). Mutants defective in chondroitin biosynthesis have a narrow or 'squashed' vulva lumen (Sqv phenotype) (*Herman et al., 1999*; *Hwang et al., 2003a*). Genetic screens for Sqv mutants identified many components of the chondroitin biosynthesis pathway (*Herman et al., 1999*; *Hwang et al., 2003a*; *Hwang et al., 2003b*), but did not identify any specific chondroitin-modified proteins, suggesting redundant contributions of multiple CPGs. To date, mass spectrometry has identified 24 CPG carrier proteins in *C. elegans*, but it is not yet known which, if any, of these contribute to vulva lumen expansion (*Noborn et al., 2018*; *Olson et al., 2006*). One of these known CPGs is FBN-1, a fibrillin-related ZP protein that is part of the worm's transient embryonic sheath matrix that precedes the cuticle (*Kelley et al., 2015*; *Labouesse, 2012*; *Noborn et al., 2018*; *Priess and Hirsh, 1986*). Other pre-cuticle aECM proteins also have been observed within the vulva and other developing tubes (*Forman-Rubinsky et al., 2017*; *Gill et al., 2016*), and the adult vulva becomes cuticle-lined in adults (*Page, 2007*; *Sulston and Horvitz, 1977*). These observations suggested to us that a transient sheath-like aECM may exist within the developing vulva lumen.

## aECM proteins show dynamic patterns of localization during vulva lumen morphogenesis

To examine aECM protein expression and localization in the vulva, we examined superfolder (Sf) GFP or mCherry-based translational fusions generated by transgenic methods or (in most cases) by CRISPR-Cas9 genome editing of the endogenous loci (Materials and methods). Six different aECM proteins were assessed (*Figure 3A*): the ZP proteins FBN-1 (*Kelley et al., 2015*), LET-653 (*Gill et al., 2016*) and NOAH-1 (*Vuong-Brender et al., 2017*), the lipocalin LPR-3 (*Forman-Rubinsky et al., 2017*), and the extracellular leucine-rich repeat only (eLRRon) proteins LET-4 (*Mancuso et al., 2012*) and SYM-1 (*Davies et al., 1999*; *Vuong-Brender et al., 2017*). We also examined a shortened version of LET-653 containing only the ZP domain, because prior studies had demonstrated that this domain is sufficient for matrix incorporation and excretory duct tube shaping (*Gill et al., 2016*; *Figure 3A*). FBN-1, NOAH-1, and LET-4 each have transmembrane domains (but could be cleaved extracellularly), while LET-653, LPR-3, and SYM-1 are secreted proteins. These fusions permitted observations of matrix structure in live L4 animals, without the matrix destruction typically induced by chemical fixation for immunofluorescence. In each case, these fusion proteins were functional as assessed by mutant rescue or strain phenotypes (Materials and methods), suggesting that their localization closely approximates the endogenous patterns. Each fusion protein initially appeared at around the L4.2 stage, and then exhibited a temporally and spatially distinct pattern over the course of L4 vulva morphogenesis (*Figure 3B–D*). All six matrix proteins were transient and disappeared in the adult stage, indicating that they are not components of the mature cuticle, but rather define a distinct matrix type present only during morphogenesis.

The ZP proteins FBN-1 and LET-653 showed somewhat complementary luminal patterns (*Figure 3B,D*). Beginning at L4.2, and as previously reported for LET-653(PAN domain) fragments (*Gill et al., 2016*), LET-653 decorated a core structure in the center of the lumen that rises to the level of the vulD and vulE cells, along with lateral elements that connect this core to the vulA, vulB1 and vulB2 cells and to the surrounding epidermis (*Figure 3B,D*). The central core structure could also be detected very weakly by DIC (*Figure 2*; red arrowhead). This core changed appearance during vulva eversion, but remained visible in the most ventral, 2°-cell-derived region through the L4.9

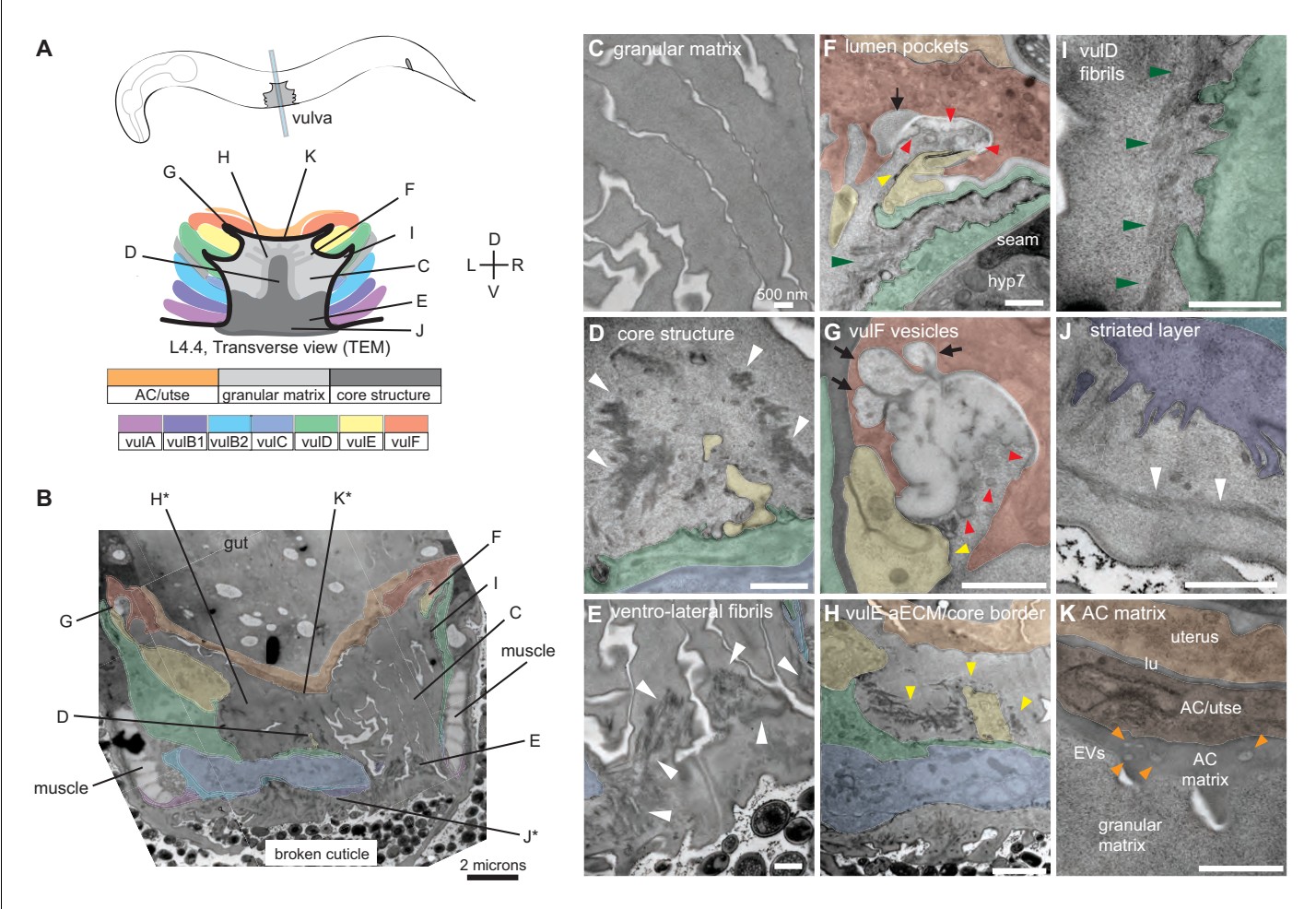

**Figure 4.** Ultrastructural features of the mid-L4 vulva aECM. (A) Transverse serial thin sections of an N2 L4.4-L4.5 stage animal were analyzed by TEM. The cartoon depicts the vulva lumen in this orientation (see also *Figure 1E*), and lines indicate the relative locations of different panel images. (B) Whole vulva view. This thin section captures a portion of the lumen and the cell borders. Vulva cells and AC/utse are pseudo-colored according to the key shown in A. Lines indicate the relative locations of different panel images, and asterisks indicate that the panel shows a region from a different thin section of the same animal. The ventral cuticle has broken during specimen processing and oval objects surrounding the specimen are *E. coli* bacteria. (C) A rough granular matrix fills the dorsal lumen. (D) Core structure (white arrowheads) rises above vulC and vulD, to the level of vulE. (E) Ventro-lateral fibrils (white arrowheads) and the ventral edge of the luminal matrix, which has pulled away from the broken cuticle. (F) The interface between vulF (red) and vulE (yellow) forms a sequestered lumen pocket where matrix accumulates (red and yellow arrowheads). vulD (green) forms another narrow lumen pocket that is densely populated with fine fibrils (green arrowhead). (G) vulF cells contain large secretory vesicles (black arrows) that are open to the extracellular space and whose contents resemble the membrane-proximal matrices that line vulF and vulE (red and yellow arrowheads, respectively). (H) Lateral view of the matrix lining vulE surfaces. (I, J) Examples of the very protrusive surfaces of 2°-derived cells (vulD and vulB1, respectively) that interface with fibrils. J also shows the ventral-most border of the aECM, which contains a striated layer (white arrowheads) similar to that seen in epidermal cuticle (*Page, 2007*). (K) A fine-grained aECM separates the AC/utse from the rougher granular matrix of the vulva lumen; this AC matrix contains numerous EVs (orange arrowheads). All scale bars are 500 nm unless otherwise indicated. See *Figure 4—figure supplement 1* for uncolored versions of all images.

The online version of this article includes the following figure supplement(s) for figure 4:

**Figure supplement 1.** Ultrastructural features of the mid-L4 vulva aECM. Uncolored images from *Figure 4* are shown.

stage. Transiently, at the L4.3-L4.5 stages, LET-653 also weakly marked the apical membranes of most cells. Finally, FBN-1 overlapped with LET-653-marked structures near vulB1 and vulB2 surfaces, but otherwise was mainly excluded from the core area and instead filled the more dorsal part of the lumen above the core (*Figure 3B,D*). During vulva eversion, FBN-1 became excluded from the dorsal-most portions of the lumen lined by 1°-derived cells, such that LET-653 and FBN-1 together

appeared to demarcate at least three separate luminal zones roughly corresponding to the regions outlined by the vulA/B cells, vulC/D cells, and vulE/F cells (*Figure 3B*).

The isolated LET-653 ZP domain and the other four aECM proteins marked specific apical membrane-proximal regions in a dynamic manner (*Figure 3C,D*). LET-653(ZP) specifically labelled just the 1°-derived vulE and vulF cell surfaces at L4.3-L4.5 stages. Previous Fluorescence Recovery After Photobleaching (FRAP) studies showed that, while it is present, this pool of LET-653(ZP) is relatively immobile, consistent with matrix incorporation (*Gill et al., 2016*). NOAH-1 faintly marked all 2° vulva cell surfaces at L4.4-L4.5, but then became increasingly concentrated on vulC and vulD. During vulva eversion, NOAH-1 prominently marked matrix spikes that protruded from vulC into the lumen, and these spikes attached to LET-653-marked lateral structures near vulB2 surfaces (*Figure 3C,D*). These NOAH-1—LET-653 connections then persisted as the lumen narrowed. LPR-3 briefly marked all vulva cell apical surfaces at early L4.4, but then became restricted to 2° cells and then specifically to vulB1 and vulB2 before largely disappearing by L4.6-L4.7 (*Figure 3D*). The departure of LPR-3 from vulC and vulD coincided with the increasingly strong presence of NOAH-1 there. The transmembrane eLRRon protein LET-4 marked all vulval apical membranes during the late L4.2-L4.7 period, and thereafter appeared intracellular (*Figure 3D*). Finally, the secreted eLRRon protein SYM-1 showed the most limited pattern, labelling vulB1 and vulB2 for just a brief period at L4.4-L4.5 (*Figure 3D*). Together, these data reveal that different combinations of aECM factors assemble on the luminal surface of each vulva cell type. Furthermore, the precise timing of each factor's appearance and disappearance points to highly regulated mechanisms for matrix assembly and remodeling.

## Ultrastructural features of the luminal matrix differ between 1° and 2°-derived vulva regions

Prior transmission electron microscopy (TEM) studies of the vulva *Gill et al., 2016*; *Herman et al., 1999* used chemical fixation methods that poorly preserved the luminal matrix and did not capture the complex luminal structures observed in the live imaging above. To obtain a clearer view of matrix ultrastructure, we turned to high pressure freezing (HPF) and freeze substitution (*Hall et al., 2012*). This method achieved much better matrix preservation and revealed many matrix layers and fibrils that we could correlate with those observed by light microscopy. Serial thin sections were collected transverse or length-wise to the body axis to obtain a three-dimensional view. Images of mid-L4 (L4.4-L4.5) and late-L4 (L4-8-L4.9) stage vulvas are shown in *Figures 4* and *5* and S1, S2. A striking feature of both stages is the difference in matrix organization in the dorsal (1°-derived) vs. ventral (2°-derived) portions of the lumen.

At the mid-L4 stage, a rough granular matrix fills the entire vulva lumen, and embedded within it are a central core structure and numerous ventro-lateral fibrillar elements similar to those seen with LET-653::SfGFP (*Figure 4B–E*). Most of the dorsal lumen surrounding the central core contains only the granular matrix; this corresponds to the region marked by FBN-1::mCherry (*Figure 3B*) and likely contains additional CPGs. There is no single luminal channel running through the granular matrix, which appears organized into multiple wide strips or flaps, each edged with a more electron-dense border (*Figure 4B,C*). 1°-derived vulval cells have relatively smooth apical surfaces lined with thin matrices that separate them from the granular matrix (*Figure 4F–H*), while the 2°-derived cells have more protrusive apical surfaces lined with numerous fibrils that are embedded within the granular matrix (*Figure 4D–F,I,J*).

At the dorsal apex of the vulva, the AC remnant hymen is lined by a layer of finer-grained, electron-dense matrix that separates it from the vulva granular matrix (*Figure 4K*). Within this AC matrix are numerous extracellular vesicles (EVs). The contents and purpose of these EVs are not currently known, but the AC is a source of multiple signaling molecules (*Hill and Sternberg, 1992*; *Sherwood and Sternberg, 2003*).

The vulF cells contain numerous large (~200 nm) secretory vesicles that resemble those seen in mammalian goblet cells (*Figure 4F,G*; *Birchenough et al., 2015*). These vesicles contain globules that resemble mucin packets, along with a few membranous intraluminal vesicles (ILVs). The secretory vesicles appear to be dumping their contents within sequestered pockets at the left and right extremes of the lumen, and these contents then expand spherically upon contacting the outside environment. The contents within these luminal pockets are continuous with a thin membrane-proximal matrix layer that likely corresponds to the layer marked by LET-653(ZP)::SfGFP (see *Figure 3D* and below). vulE surfaces are decorated by a mesh-like matrix that drapes down along the top

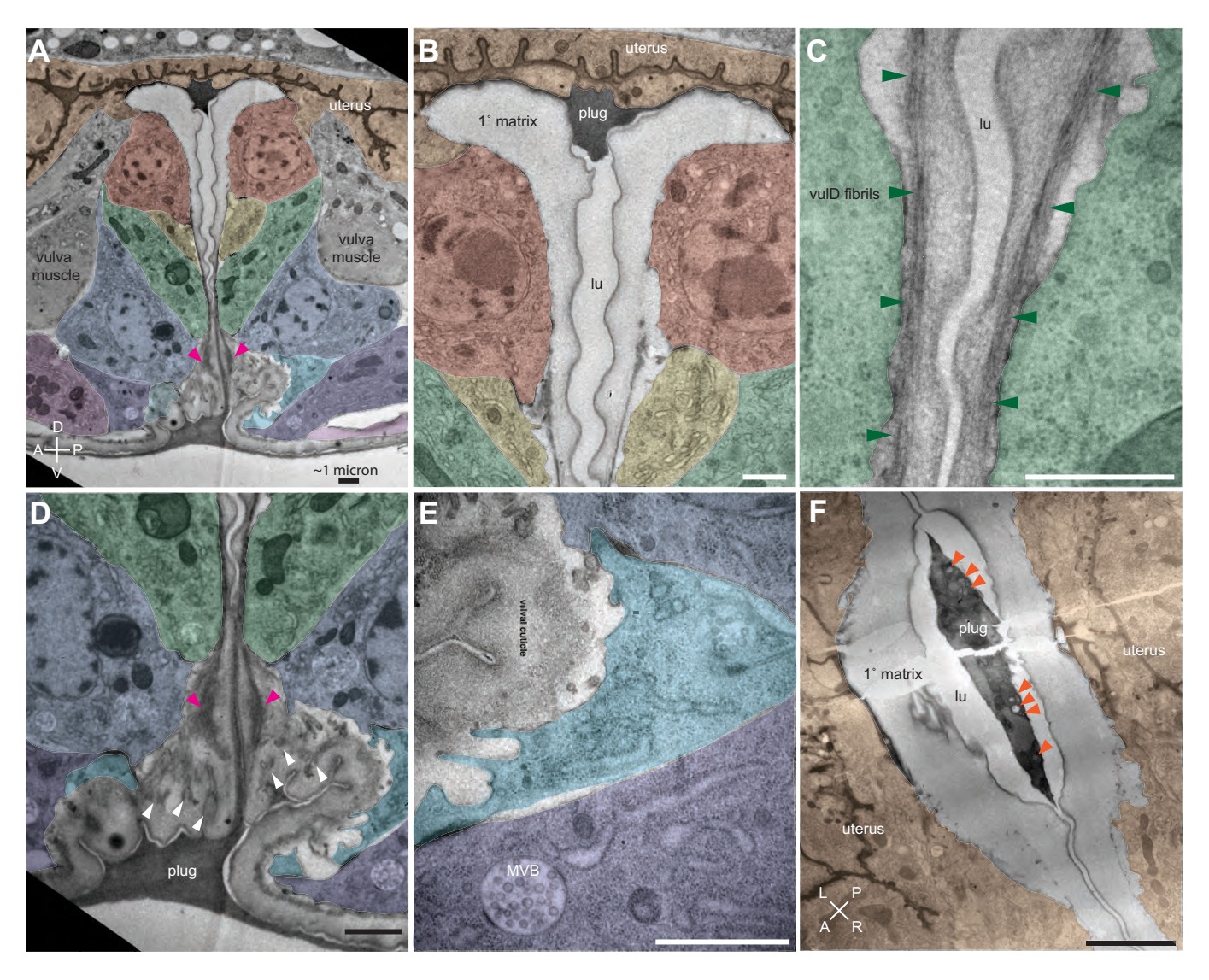

**Figure 5.** Ultrastructural features of the late L4 vulva aECM. (A) Longitudinal slice through the vulva of an N2 L4.8-L4.9 stage animal, with orientation similar to that in confocal images. Vulva and uterine cells are pseudo-colored as in *Figure 4B*. Pink arrowheads indicate matrix spikes as observed with NOAH-1::mCherry (see *Figure 3C*). (B–E) Higher magnification views of the specimen in panel A. (B) Primary vulva cells are covered in a thick membrane-proximal matrix, and the dorsal-most edge of the lumen is filled with a plug of darkly-staining material. (C) The membrane-proximal matrix continues over vulD and vulC, but becomes filled with dense fibrils (green arrowheads). (D) Matrix spikes (pink arrowheads) extend from vulC/D into a cuticle-like matrix below. Various other fibrils (white arrowheads) are present within this ventral matrix. (E) Multi-layered nascent cuticle over vulB2. Note protrusive surface of vulB2 and multi-vesicular body (MVB) within vulB1. Many MVBs are present in vulva cells at this stage. (F) Longitudinal dorsoventral slice through uterine cells and the primary vulva matrix and plug of a second N2 L4.8-L4.9 stage animal. Note the numerous EVs (orange arrowheads) present within the plug. All scale bars, one micron. See *Figure 5—figure supplement 1* for uncolored versions of all images.

The online version of this article includes the following figure supplement(s) for figure 5:

**Figure supplement 1.** Ultrastructural features of the late L4 vulva aECM. Uncolored images from *Figure 5* are shown.

border of the core and ventro-lateral fibrils (*Figure 4H*). This matrix appears as dark membrane-associated patches when vulE is viewed in cross-section (*Figure 4F,G*). This matrix may serve as the barrier that excludes FBN-1 from the ventral fibrillar region (see *Figure 3B,D*).

vulD and vulC surfaces that sit above the ventro-lateral fibrils (and external to the core) are lined with thin fibrils that run in a dorsal-ventral orientation, parallel to the cell membranes (*Figure 4I*). These fibrils are embedded within the granular luminal matrix rather than forming a separate layer,

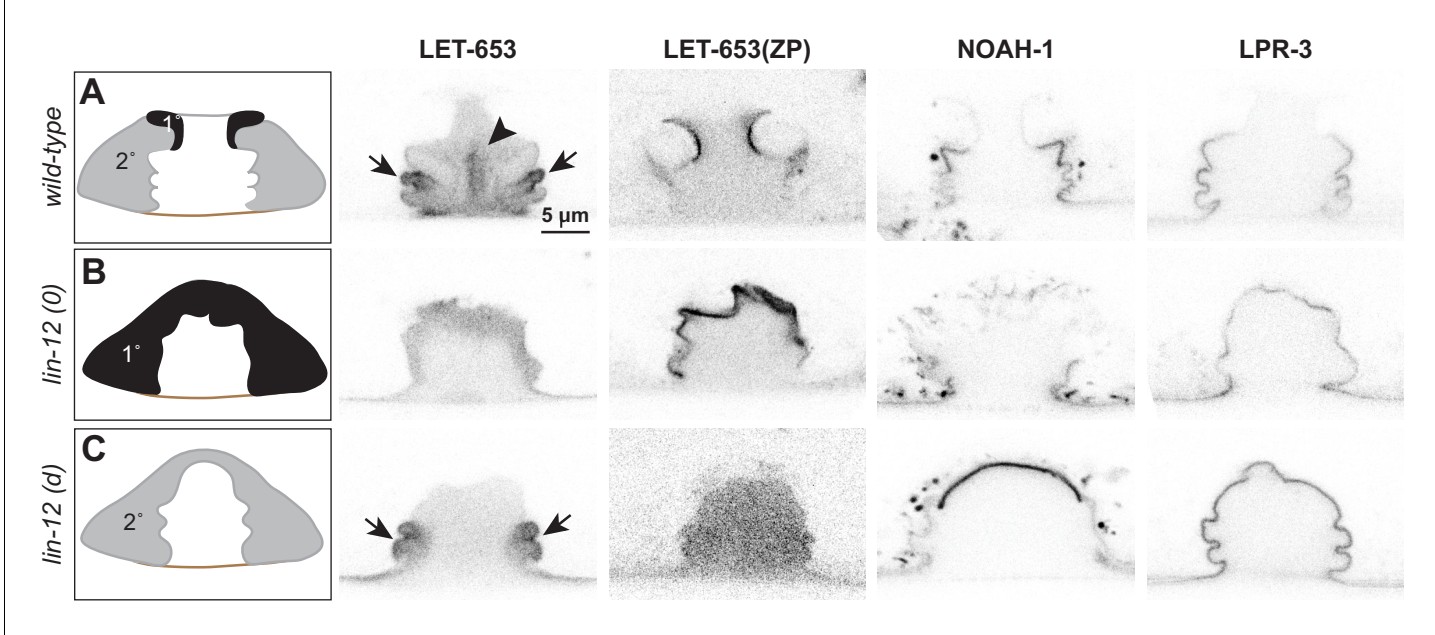

**Figure 6.** Different vulva cell types produce and assemble different aECMs. (A–C) Panels in left column show cartoons of vulva cell types and lumen shape at mid-L4. Remaining columns show single confocal slices through the vulva lumen. Fusions used are LET-653(full-length)::SfGFP (*cs262*), LET-653 (ZP)::SfGFP (*csIs66*), NOAH-1::mCherry (*mc68*), SfGFP::LPR-3 (*cs250*). At least n = 8 L4s were imaged for each strain. (A) In wild-type animals, full-length LET-653 predominantly labels the core and ventro-lateral fibrils, LET-653(ZP) labels the membrane-proximal matrix over 1° cells, NOAH-1 labels membrane-proximal matrices over 2° cells (especially vulC and vulD), and LPR-3 transiently labels membrane-proximal matrices over all cells, but then becomes concentrated over 2° cells (see also *Figure 3D*). (B) Loss of 1° cells in *lin-12(0)* (null, *n137n720*) mutants disrupted the luminal core and NOAH-1 localization. Most of the NOAH-1 pattern here is intracellular. (C) Loss of 2° cells in *lin-12(d)* (hypermorphic, *n137*) mutants disrupted the luminal core and LET-653(ZP) localization.

and they abut numerous small cellular projections. The fibrils are particularly concentrated in narrow (~0.5 micron) lumen pockets generated by the complex shape of vulD (*Figure 4F*). These fibrillar regions correspond to those surfaces that become strongly marked by NOAH-1::mCherry (see *Figure 3C*).

Finally, the ventral-most vulC surfaces, as well as vulB2, vulB1 and vulA, interface with the dense ventro-lateral fibrils, which run both perpendicular to and parallel with the cell membranes (*Figure 4B,E*). The cell surfaces that interface with these fibrils are extremely protrusive (*Figure 4J*). At the most ventral edge of the lumen, beneath the core fibrils, matrix layers that resemble those of the epidermal sheath and nascent cuticle interface with the remaining L4 epidermal cuticle (which has broken and pulled away somewhat in this specimen) (*Figure 4B,E,J*).

The late-L4 stage vulva (*Figure 5*) retains several of the matrix features seen at the earlier stage, with some notable differences. An electron-dense 'plug' resembling the earlier AC matrix is present at the dorsal apex and contains EVs (*Figure 5A,B,F*). A similar-appearing matrix is also present at the ventral opening of the lumen (*Figure 5A,D*). The 1°-derived vulE and vulF cells and the 2°-derived vulC and vulD cells are now covered with a thick membrane-proximal matrix that somewhat resembles the prior CPG matrix, but with a well-defined, darkly-staining border and a single open channel that runs through its center (*Figure 5A,B,F*). The membrane-proximal matrix over vulC and vulD also contains many dense fibrils that extend down into the matrix below (*Figure 5C,D*); these likely correspond to the NOAH-1-marked matrix spikes observed by confocal imaging (see *Figure 3C*). The remaining vulA, vulB1 and vulB2 cells are covered with a more complex, multi-layered matrix that resembles the nascent pre-cuticle on nearby epidermal cells (*Figure 5D,E*). Various fibrillar structures are embedded within this thick matrix (*Figure 5D*), possibly corresponding to the core and ventro-lateral fibrils seen earlier. Thus, just as seen by confocal light microscopy at this stage (*Figure 3B*), TEM shows three distinct luminal zones corresponding to the regions outlined by the vulA/B cells, vulC/D cells, and vulE/F cells.

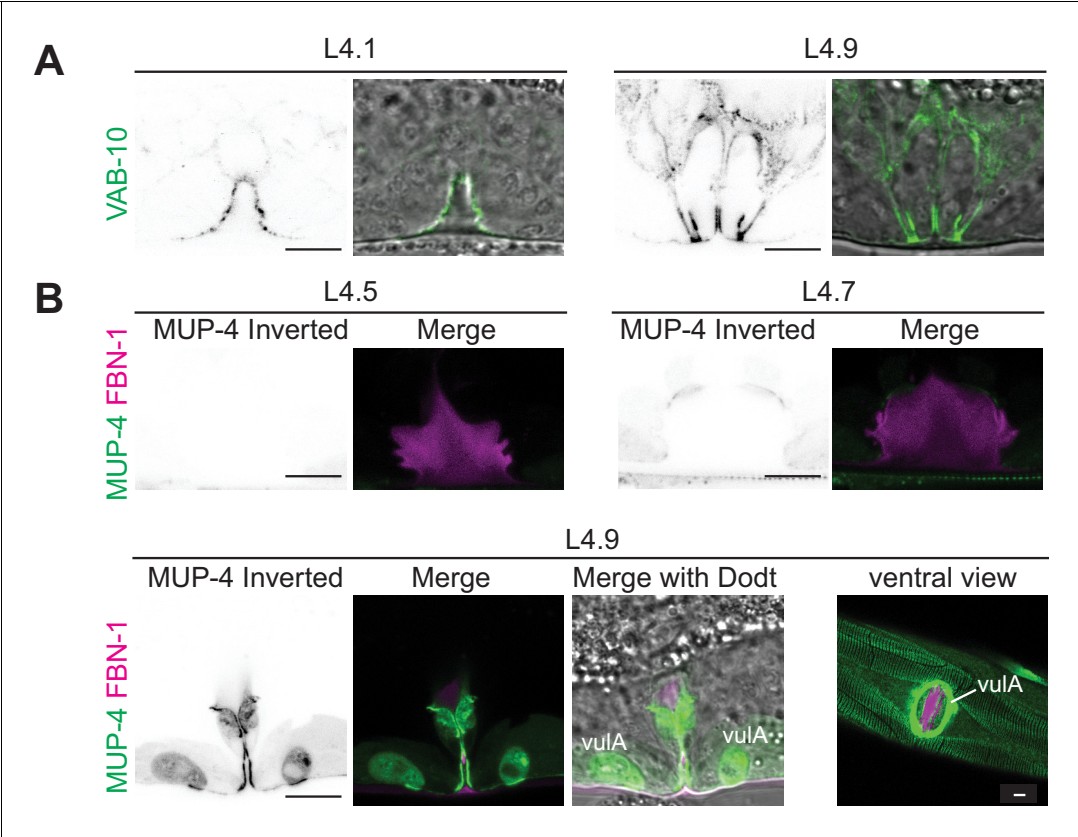

**Figure 7.** Vulva aECM assembles prior to expression of MUP-4/matrillin. (**A**) VAB-10::GFP (*cas627*) marked all apical membranes in the vulva throughout L4 (including 4/4 L4.1/L4.2 stage animals). (**B**) MUP-4::GFP (*upIs1*) marked apical membranes beginning in late L4 (0/3 L4.4/L4.5, 3/3 L4.6/L4.7, 2/2 L4.9). FBN-1::mCherry (*aaaIs12* or *aaaEx78*) is also shown. At L4.9, MUP-4 expression is particularly strong in the vulA toroid, which surrounds the vulva opening and connects to the surrounding hyp7 epidermis. The remaining FBN-1 matrix connects to vulA at the left and right sides of the lumen, as seen in the ventral view.

## 1° and 2° vulva cell types produce and assemble different matrices

To better understand the differences between the matrix produced and assembled by 1° vs. 2° vulva cell types, we analyzed matrix patterns in *lin-12/Notch* mutants. LIN-12/Notch promotes 2° vs. 1° VPC fates, so loss-of-function [*lin-12(0)*] mutants have only 1° vulva cell types, while gain-of-function [*lin-12(d)*] mutants have only 2° vulva cell types (*Greenwald et al., 1983*; *Sternberg and Horvitz, 1989*). *lin-12(0)* and *lin-12(d)* mutants both have well-inflated (though mis-shaped) vulva lumens (*Figure 6*), indicating that relevant CPGs are made by both sets of vulva cell types. Indeed, *Herman et al., 1999* previously showed that both types of *lin-12* mutants require the Sqv chondroitin biosynthesis pathway for lumen inflation.

Close examination of *lin-12* mutants suggested, however, that the central core structure was missing. When LET-653::SfGFP was introduced into *lin-12(0)* mutants, core structures appeared very meager or absent (*Figure 6B*). In *lin-12(d)* mutants, no core was observed in the central lumen, but some ventro-lateral elements were still present at the vulB 'fingers' (*Figure 6C*). We conclude that both 1° and 2° cells are required to generate the core, but that 2°-derived cells (most likely vulB1 and vulB2) generate at least some of the ventro-lateral elements independently.

*lin-12* mutants also showed changes in the localization of membrane-proximal matrix factors, as predicted based on the cell fate changes. For example, LET-653(ZP) marked all vulva apical membranes in *lin-12(0)* mutants, but none in *lin-12(d)* mutants (*Figure 6*). Since *let-653* is expressed by all seven vulva cell types (*Gill et al., 2016*), these data suggest that a 1°-specific partner is required to recruit LET-653(ZP) to the membrane-proximal matrix. In contrast, NOAH-1 was mostly absent from vulva apical membranes in *lin-12(0)* mutants, but strongly marked the dorsal-most apical membranes in *lin-12(d)* mutants (*Figure 6*). LPR-3 marked some apical membranes in both *lin-12(0)* and *lin-12(d)*

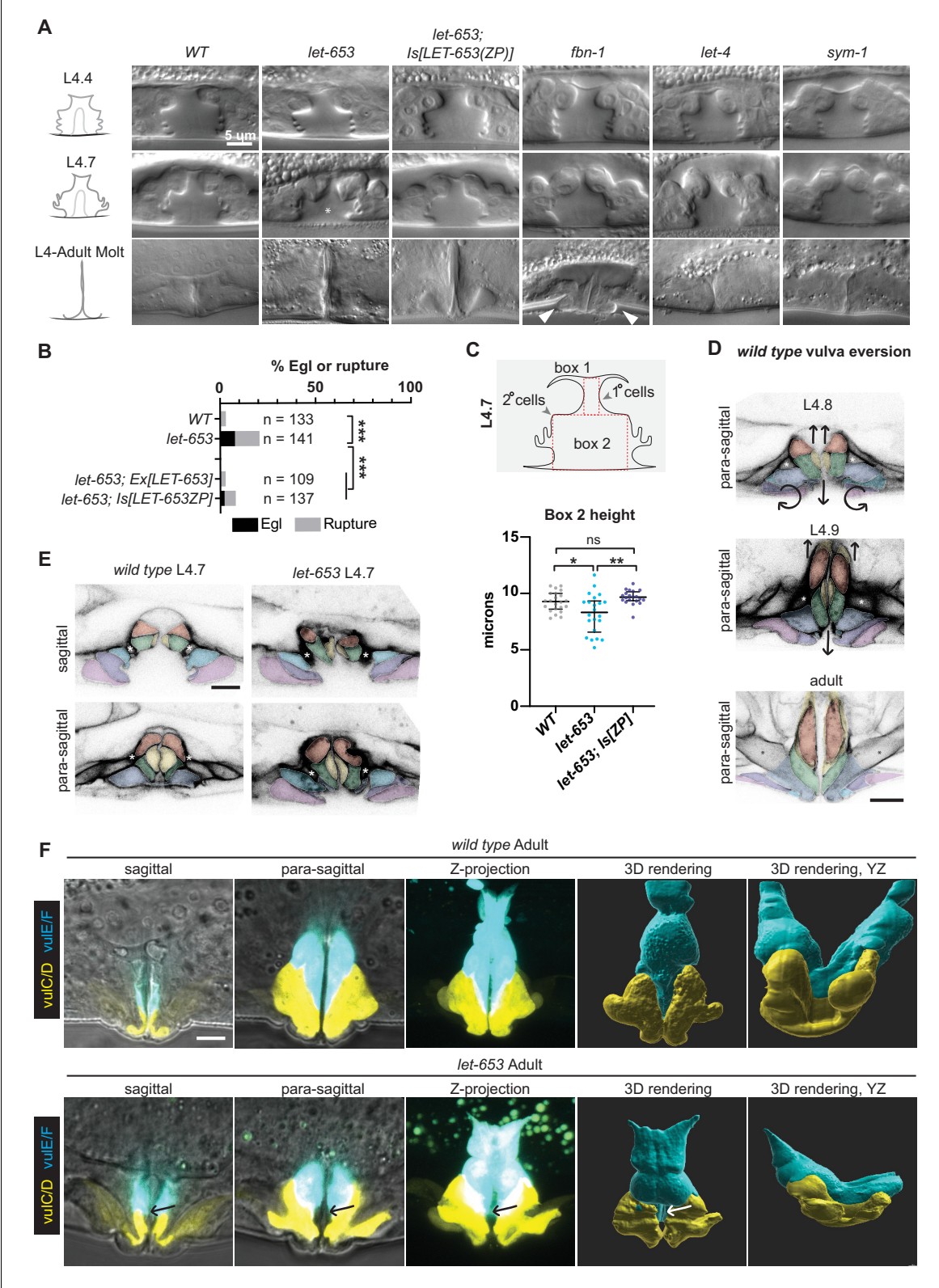

**Figure 8.** Individual aECM factors play subtle roles in vulva eversion. (**A**) DIC images of mutant vulvas at L4.4, L4.7 and L4.9-adult molt. At least 40 L4 animals of each genotype were imaged, including at least five each of the three stages shown. Alleles used: *fbn-1(tm290)*, *let-653(cs178)* (strains UP3342 and UP3422), *let-4(mn105)*, and *sym-1(mn601)*. Asterisk indicates collapsed lumen morphology in some *let-653* mutants (n = 6/22 L4.7, see panel C). Arrowheads indicate abnormal bulges of the vulA and vulB1/B2 cells in *fbn-1* mutants (n = 9/13 L4.9). (**B**) A small proportion of *let-653* mutants had

*Figure 8 continued on next page*

*Figure 8 continued*

progeny that hatched in utero (Egl phenotype) or ruptured at the vulva within eight days of reaching adulthood. These phenotypes were rescued by transgenes expressing full-length LET-653 or just the ZP domain. ***p<0.0001, Fisher's exact test. (C) *Let-653* mutants have reduced lumen dimensions at the L4.7 stage (n = 22). To measure lumen dimensions, the largest box possible was drawn within the 1°-generated (Box 1) and 2°-generated (Box 2) lumen spaces, as visualized by DIC. Dimensions for Box 1 did not differ significantly between genotypes (*Figure 8—figure supplement 1*), but Box 2 height was somewhat reduced. This phenotype was rescued by a transgene expressing the LET-653 ZP domain. *p=0.031, **p=0.001, WT vs. *let-653;Is [ZP]* p=0.085, Mann–Whitney U test. (D) *WT* vulva eversion. Membranes were visualized with MIG-2::GFP. Compare these para-sagittal slices to sagittal slices of the same animals in *Figure 2*. n = 3 per stage. (E) *let-653* mutants have irregular vulva cell shapes at the onset of vulva eversion (L4.7 stage). Membranes were visualized with MIG-2::GFP. Asterisks indicate the vm1 sex muscles. Both sagittal and para-sagittal slices from confocal Z-stacks are shown. In wild-type, vulva cells are symmetrical across the midline, but in *let-653* mutants, cell shape and position are mismatched (n = 3/3; *WT*: n = 0/6). No defects in vulva cell fusion were observed (*Figure 8—figure supplement 2*). (F) *let-653* mutants have subtly irregular vulva cell shapes as older L4s and adults. vulE and vulF were visualized with *daf-6pro::CFP*, and vulC and vulD were visualized with *egl-17pro::CFP* (*Mok et al., 2015*). Both sagittal slices and Z-projections from confocal Z-stacks are shown, along with three- dimensional renderings generated with Imaris (Bitplane) from those Z-stacks. In *let-653* mutants, cell shape and position are variably abnormal (n = 4/4, *WT*: 0/3). In the specimen picture, vulE/F appear less elongated along the dorsal-ventral axis and vulC/D are slightly flattened relative to *WT*. Arrows indicate the transition zone between vulE/F and vulC/D, where an abnormal gap occurs in the mutant. In the YZ view, *WT* primary cells form a deep U-shape as they extend toward the seam cells, but *let-653* primary cells form a much shallower curve.

The online version of this article includes the following source data and figure supplement(s) for figure 8:

**Source data 1.** Percentage of *WT* and mutants *C. elegans* that fail to lay eggs.
**Source data 2.** Measurements of *WT* and mutant L4.7 stage vulvas.
**Figure supplement 1.** Measurements of *let-653, mig-22,* and LET-653+ vulvas Lumen dimensions at the L4.4 and L4.7 stages were quantified as in *Figures 8* and *11*.
**Figure supplement 2.** *let-653* mutant vulvas have normal cell fusion L4 4 stage *WT* and *let-653(cs178)* vulvas with the apical junction marker DLG-1::GFP (*mc103*).

---

mutants, but was more robust in the latter, consistent with its 1° and 2° membrane-binding patterns in wild-type (*Figure 6*). In summary, the results of these experiments indicate that each cell's identity, rather than its position in the organ, determines what matrix factors assemble on its surface.

## MUP-4/matrillin expression coincides with appearance of mature cuticle

The continuous movement of cells and the changes in aECM composition during vulva morphogenesis, including the eventual transition from pre-cuticle aECM to cuticle, suggest dynamic cell-cell and cell-aECM attachments. If or how the above aECM proteins attach to apical membranes remain unknown, particularly since most either do not have transmembrane domains (*Figure 3A*), do not require their transmembrane domains for function (*Mancuso et al., 2012*), or are likely cleaved to release the extracellular domains from their transmembrane domains (*Bokhove and Jovine, 2018*). Body cuticle attaches to cell surfaces via hemi-desmosome-like structures (CeHDs) and fibrous organelles (FOs) that span the epidermis and link to body muscle (*Pasti and Labouesse, 2014*). Apically, the CeHD component VAB-10a (related to spectraplakin) (*Bosher et al., 2003*) links to the matrillin-like transmembrane proteins MUP-4 (*Hong et al., 2001*) and MUA-3 (*Bercher et al., 2001*) to connect the epidermis and cuticle (*Suman et al., 2019*). However, it is not known whether similar complexes also link other types of aECMs to epithelial cells, particularly in regions where body muscle is not adjacent.

To address whether the known matrix-anchoring complexes could anchor aECMs in the vulva, we asked where VAB-10a and MUP-4 appear in vulval cells. VAB-10::GFP was present along apical membranes of all vulva cells throughout L4 morphogenesis (*Figure 7A*). However, MUP-4::GFP appeared only later, at the L4.6-L4.7 stage, and was enriched over vulC and vulD, where the vm1 sex muscles attach (*Figure 7B*). By L4.9 and adult, when cuticle is present, MUP-4::GFP lined all vulva apical membranes, but it was expressed most prominently in the vulA toroid, which links the vulva to the surrounding epidermis and body muscle (*Figure 7B*). Therefore, while MUP-4 and VAB-10 could potentially help connect the cuticle to vulva cells just as they do in the epidermis, different (possibly also VAB-10-affiliated) linkers likely recruit the earlier pre-cuticle aECM factors. Consistent with this hypothesis, prior large-scale RNAi studies revealed protruding vulva defects (indicative of abnormal eversion) after depletion of *vab-10*, but not after depletion of *mup-4* or *mua-3* (*Shephard et al., 2011*; *Simmer et al., 2003*). Further experiments will be needed to test the roles

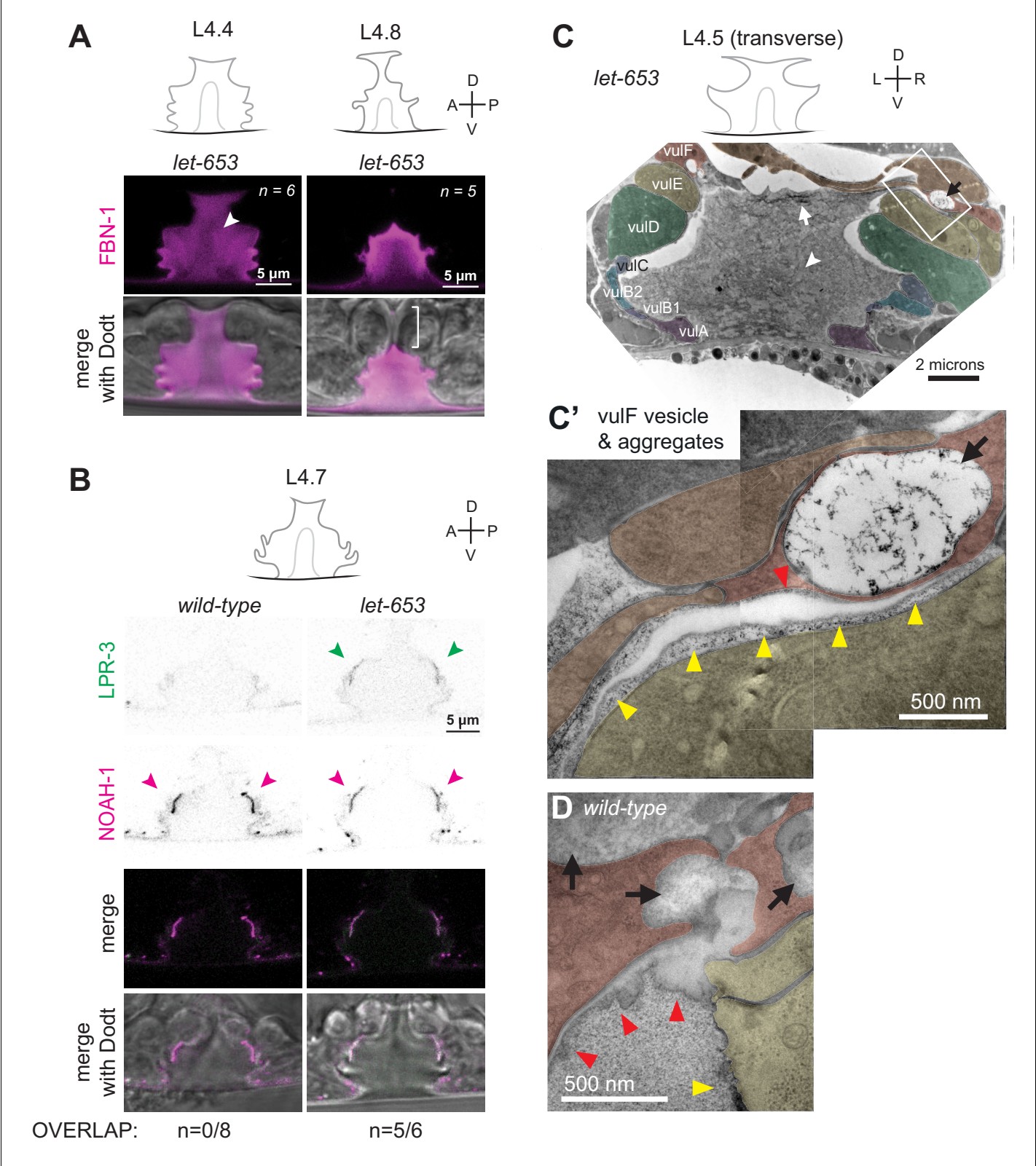

**Figure 9.** The ZP protein LET-653 is required for proper organization and remodeling of the vulva aECM. (A) *Let-653(cs178)* mutants showed relatively normal patterns of FBN-1::mCherry (*aaaIs12*) localization. Arrowhead indicates exclusion of FBN-1 from the core region. Bracket indicates exclusion of FBN-1 from the 1° lumen during eversion (n = 5/5). Compare to *WT* in **Figure 3B**. (B) *Let-653(cs178)* mutants showed normal recruitment of SfGFP::LPR-3 (*cs250*) and NOAH-1::mCherry (*mc68*) to 2° surfaces, but abnormally delayed clearance of SfGFP::LPR-3 from vulC/D. p=0.0445, Fisher's exact test. (C) *Figure 9 continued on next page*

Figure 9 continued

Transverse TEM slice of a *let-653(cs178)* mutant at mid-L4 (~L4.5) stage. Compare to the *WT* mid-L4 specimen in *Figure 4B*. Fibrils are present near the AC/utse (arrow) and the core structure (arrowhead) is not well-defined, unlike in *WT* (see magnified images in *Figure 9—figure supplement 1*). Box indicates region magnified in C'. (**C'**) An abnormal secretory vesicle in vulF is filled with dark aggregates that match those present in the membrane-proximal matrix over vulF (red arrowhead) and vulE (yellow arrowheads). A similar matrix continues beneath the AC/utse (see also *Figure 9—figure supplement 1*). (**D**) *WT* vulF vesicles and matrix for comparison. See also *Figure 4G*.

The online version of this article includes the following figure supplement(s) for figure 9:

**Figure supplement 1.** Vulva aECM organization differs between *let-653* mutants and *WT*.

of VAB-10, MUP-4 and MUA-3 more definitively, and to identify the mechanisms that link aECM to vulva apical membranes.

## Transient aECM factors facilitate proper vulva eversion

The complex localization patterns described above suggest important roles for aECM factors in vulva morphogenesis. Of the six aECM factors described here, only *sym-1* mutants are fully viable, while presumed null mutants of the rest mostly arrest as L1 larvae with excretory tube blockage or other epithelial tissue-shaping defects (*Forman-Rubinsky et al., 2017*; *Gill et al., 2016*; *Mancuso et al., 2012*; *Pu et al., 2017*; *Soulavie et al., 2018*; *Vuong-Brender et al., 2017*). To examine vulva phenotypes in these lethal mutants, we took advantage of rare escapers (for *fbn-1*) or used tissue-specific rescue strains (Materials and Methods) to bypass the earlier requirements (for *let-653* and *let-4*). We were not able to examine *noah-1* or *lpr-3* mutants in this study because of their severe epidermal molting defects (*Forman-Rubinsky et al., 2017*; *Vuong-Brender et al., 2017*). All mutants examined had fairly normal vulva lumens at the mid-L4 stage, indicating efficient CPG-dependent lumen inflation (*Gill et al., 2016*; *Figure 8A*). All mutants also assembled a core structure, as seen by DIC (*Figure 8A*). However, in some cases, later stage vulvas appeared mis-shapen or improperly everted (*Figure 8A*). Specifically, some *let-653* mutants had a prematurely collapsed vulva lumen (*Figure 8A*, asterisk), and most *fbn-1* mutants had abnormal bulges of the outer vulA, vulB1 and vulB2 cells (*Figure 8A*, arrowheads). We conclude that aECM factors facilitate later stages of vulva morphogenesis, including vulva eversion.

To better understand these phenotypes, we focused on *let-653* mutants. As adults, most *let-653* mutants were able to lay eggs; however, a small proportion of mutants ruptured at the vulva and/or were egg-laying defective (Egl) (*Figure 8B*). Measurements of the 1°- and 2°-derived portions of the vulva lumen confirmed normal dimensions at the L4.4 stage, but slightly reduced dimensions in a subset of animals at the onset of eversion at the L4.7 stage (*Figure 8C*, *Figure 8—figure supplement 1*). A transgene expressing just the LET-653(ZP) domain reversed the L4.7 defects, and led to overexpansion of the lumen at L4.4 (*Figure 8C*, *Figure 8—figure supplement 1*), consistent with our prior report (*Gill et al., 2016*) that this domain has lumen-expanding properties.

Vulva eversion has been described as the vulva 'turning inside out' (*Seydoux et al., 1993*; *Sharma-Kishore et al., 1999*), but the specific cellular events involved have never been reported. To visualize cell positions and shapes during eversion, we used the RhoG marker MIG-2::GFP to label all vulva cell membranes (*Figures 2* and *8D,E*), and the *daf-6pro::CFP* and *egl-17pro::YFP* marker combination (*Mok et al., 2015*) to label the vulE/F and vulC/D cells specifically (*Figure 8F*). During wild-type eversion (*Figure 8D*), all four of these cells elongate in the dorsal-ventral axis to partly occlude the luminal space. vulC extends a narrow NOAH-1 matrix spike into the core matrix, whose spokes appear to fold like those of an umbrella as the lumen narrows (*Figure 3C*). The vm1 sex muscles also extend ventrally into vulC and vulD (*Figures 2* and *8D*). Meanwhile, vulA, vulB1 and vulB2 tilt ventrally, while vulE reaches dorsally to connect to the seam epidermis. By adulthood, vulE and vulF enclose the bulk of the lumen, vulC and vulD form the vulva lips, and the vulA, vulB1 and vulB2 cells are excluded from the lumen, and instead form the epidermis surrounding the vulva opening (*Figures 2* and *8D,F*).

In *let-653* mutants, vulva eversion occurred in a more irregular manner (*Figure 8*). The anterior and posterior halves of the vulva showed various asymmetries by the onset of eversion at the L4.7 stage (*Figure 8E*), and cell shapes and relative cell positions continued to be variably abnormal in older L4s and adults (*Figure 8F*). Defects were particularly noticeable in para-sagittal slices and in

3D reconstructions of the YZ dimensions; whereas *WT* adult vulE cells reach far dorsally to connect to the lateral seam epidermis on the left and right sides of the body (*Sharma-Kishore et al., 1999*), some *let-653* vulE cells appeared not to reach that far (*Figure 8F*). Imaging of a cell junction marker revealed that *let-653* morphogenesis defects are not due to failure of cell fusion to form the vulva toroids (*Figure 8—figure supplement 2*). Instead, we conclude that LET-653 plays subtle roles to coordinate the complex cell shape changes that occur during the process of eversion.

## LET-653 is required for multiple aspects of vulva aECM organization

To test how LET-653 affects the organization of the vulva matrix, we first assessed the status of other matrix factors in the *let-653* mutant background. *let-653* mutants still assembled some type of central core structure, as seen by DIC (*Figure 8A*) and by the exclusion of FBN-1 from this region (*Figure 9A*). As in *WT*, FBN-1 departed from the dorsal-most lumen during eversion (*Figure 9A*) and LPR-3 and NOAH-1 still appeared at their proper locations (*Figure 9B*). However, the normally precise sequence of LPR-3 clearance was disrupted, such that LPR-3 remained on vulC and vulD apical surfaces longer than normal, and overlapped significantly with NOAH-1 there (*Figure 9B*), suggesting improper organization of the vulC/D membrane-proximal aECM.

TEM of a mid-L4 *let-653* mutant revealed more dramatic matrix abnormalities (*Figure 9C–C′*). Some vulF secretory vesicles contained very disorganized, dark aggregates, and the membrane-proximal matrices over vulF, vulE, and the AC/utse appeared filled with such aggregates (*Figure 9C′*). The luminal matrix also contained many aggregates or short, fibrillar structures rather than having the uniform granular appearance of *WT*. Few fibrils were detected near the surfaces of vulC/D or other 2°-derived cell types. Instead, many long fibrils accumulated at the dorsal apex of the vulva, along the AC/utse, where fibrils had not been seen in *WT* (*Figure 9D*, *Figure 4—figure supplement 1*). The central core region was recognizable but less well-defined than in *WT*, as it was interspersed with many aggregates or short fibrils similar to those present in the rest of the luminal matrix (*Figure 9C*, *Figure 4—figure supplement 1*). Thus, although *let-653* is not required to build the luminal core or to establish membrane-proximal matrices per se, it is required for the proper morphology of these structures and to set up the major dorsal vs. ventral differences in the granular vs. fibrillar organization of the vulva aECM.

## Vulva aECM structures form independently of chondroitin

Thus far, the data indicate that chondroitin plays an early role in vulva lumen inflation, whereas LET-653 and FBN-1 play later roles in morphogenesis and eversion. However, FBN-1 is a known CPG (*Noborn et al., 2018*) and other aECM factors appear to be embedded within the CPG matrix (*Figure 4*), suggesting some coordination between the two processes.

To ask if chondroitin affects aECM assembly, we examined aECM reporters in *sqv-5* and *mig-22* mutants (*Figure 10*). *sqv-5* and *mig-22* encode chondroitin sulfate synthases related to human CHSY1 and CHSY2 (*Hwang et al., 2003a*; *Suzuki et al., 2006*), which promote chondroitin biosynthesis and polymerization (*Izumikawa et al., 2004*). LET-653(ZP) did still assemble on the apical surfaces of 1° cells in *sqv-5* mutants (*Figure 10A*). However, these null mutants have a very narrow vulva lumen, which made it difficult to stage L4 animals accurately and assess the localization of full-length LET-653 or other dynamic aECM factors. Therefore, we turned to *mig-22* hypomorphic (rf = reduced function) mutants, which have a slightly less severe vulva phenotype.

In *mig-22(rf)* mutants at mid-L4 stage, both the 1°- and 2°-derived parts of the lumen were narrower than in wild-type, but lumen length varied between regions (*Figures 10B,C* and *11A,B*). The vulva 'neck' region, defined by vulD, vulE, and vulF (see *Figure 2*), was longer than in wild-type, and vulD cells were prematurely elongated along the dorsal-ventral axis (*Figure 10B*), as previously described for *sqv-3* mutants (*Herman et al., 1999*). In contrast, the main body of the lumen, defined by vulA, vulB1, vulB2, and vulC, was shorter. Thus chondroitin has very different effects on the apical domain size of different cell types.

*mig-22(rf)* mutants still had a luminal core structure marked by LET-653::SfGFP, though this was narrower than in wild-type, matching the changed dimensions of the lumen (*Figure 10D*). *mig-22(rf)* mutants also still recruited LET-653(ZP), NOAH-1 and LPR-3 to appropriate apical surfaces (*Figure 10A,E*). Thus, chondroitin does not appear essential for assembling the vulva aECM, though it remains possible that it influences aECM structure in a more subtle way.

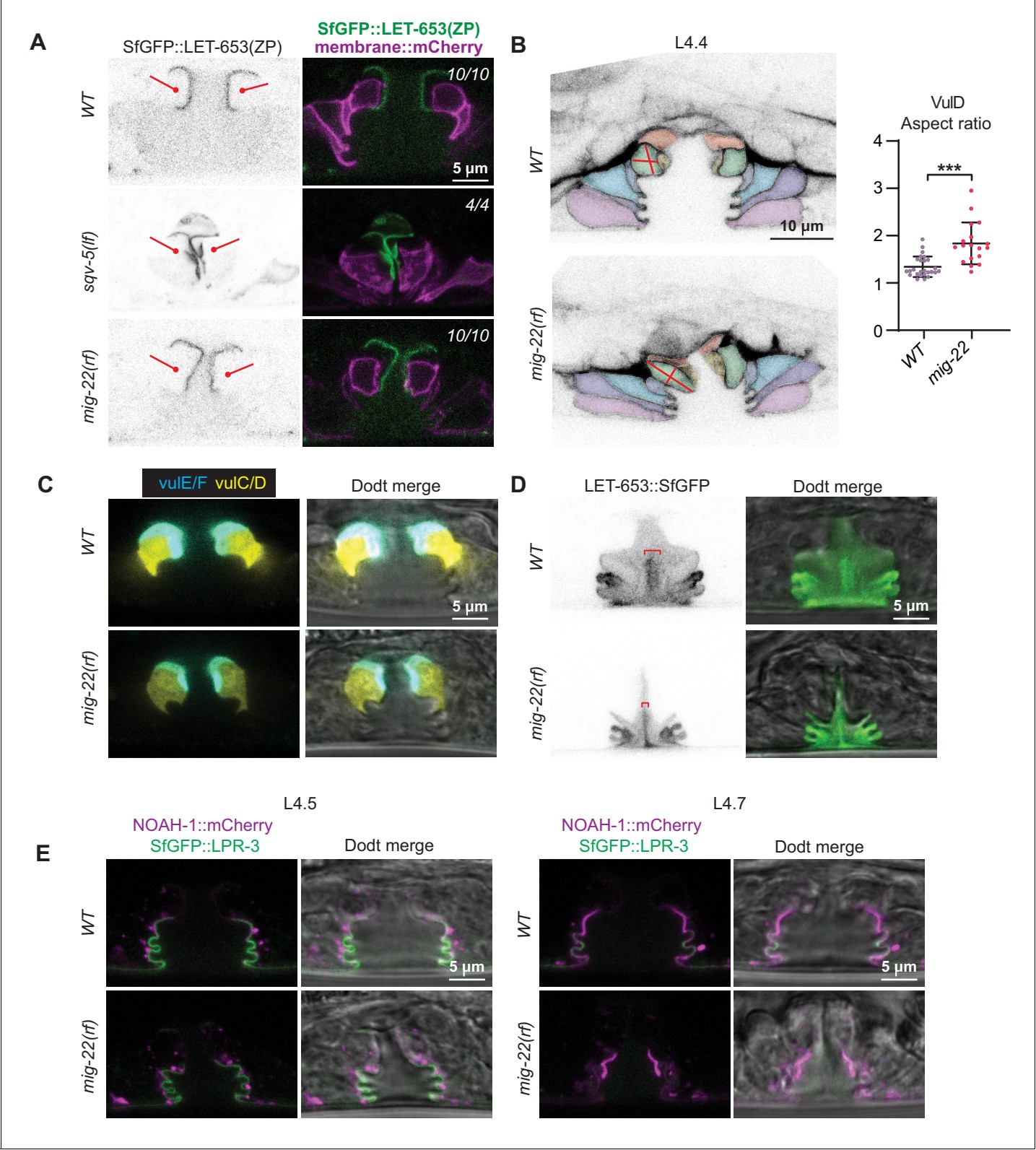

**Figure 10.** Vulva aECM structures form independently of chondroitin. (**A**) Chondroitin mutants showed normal recruitment of LET-653(ZP)::SfGFP (*csIs66*) to the membrane-proximal matrix over 1° cells. Alleles used were *sqv-5(n3611)* (n = 4) and *mig-22(k141rf)* (n = 10). (**B**) WT vs. *mig-22(k141rf)* mutants (L4.4 stage) with membrane marker MIG-2::GFP (*muIs28*). The vulva neck region was taller and narrower in mutants compared to wild- type, while the rest of the lumen was shorter and narrower (n = 15, see quantification in **Figure 11B**). vulD cells (green) showed the most dramatic shape

*Figure 10 continued on next page*

Figure 10 continued

changes, with an increased aspect ratio (longest axis/shortest axis, p<0.001, Mann Whitney two tailed U test). *WT*, n = 26. *mig-22*, n = 18. (C) WT vs. *mig-2(k141rf)* mutants (L4.5 stage) with vulE/F marker *daf-6pro::CFP* and vulC/D marker *egl-17pro::YFP* (n = 3). (D) *mig-22(k141rf)* mutants assembled a well-organized, but narrow, core structure, as seen with LET-653::SfGFP (*cs262*) (n = 8). (E) *mig-2(k141rf)* mutants showed normal localization of LPR-3::SfGFP and NOAH-1::mCherry to apical surfaces (n = 10).

The online version of this article includes the following source data for figure 10:

**Source data 1.** Shape description of *WT* and *mig-22* vulD cells.

## Chondroitin and the vulva aECM have both lumen-expanding and lumen-constraining properties

To ask if chondroitin and aECM factors work cooperatively to shape the vulva, we examined double mutants between *mig-22(rf)* and *let-653*. Surprisingly, loss of *let-653* largely suppressed the *mig-22 (rf)* mutant Sqv phenotype. At the L4.4 stage, double mutants appeared properly inflated in the ventral region, and actually overly inflated in the dorsal, 1°-derived region (*Figure 11A,B*). Nevertheless, at later eversion stages, the vulva lumen appeared variably abnormal and contained disorganized matrix material (*Figure 11C*), and as adults, almost all double mutants were Egl or ruptured at the vulva (*Figure 11D*). These results indicate that MIG-22 and LET-653 have opposing roles in promoting vs. restraining initial lumen inflation, but also have cooperative roles in restraining the subsequent expansion of the dorsal-most 1° vulva toroids, and in promoting later steps of eversion and cuticle formation. These results are inconsistent with the model that chondroitin acts only to exert a

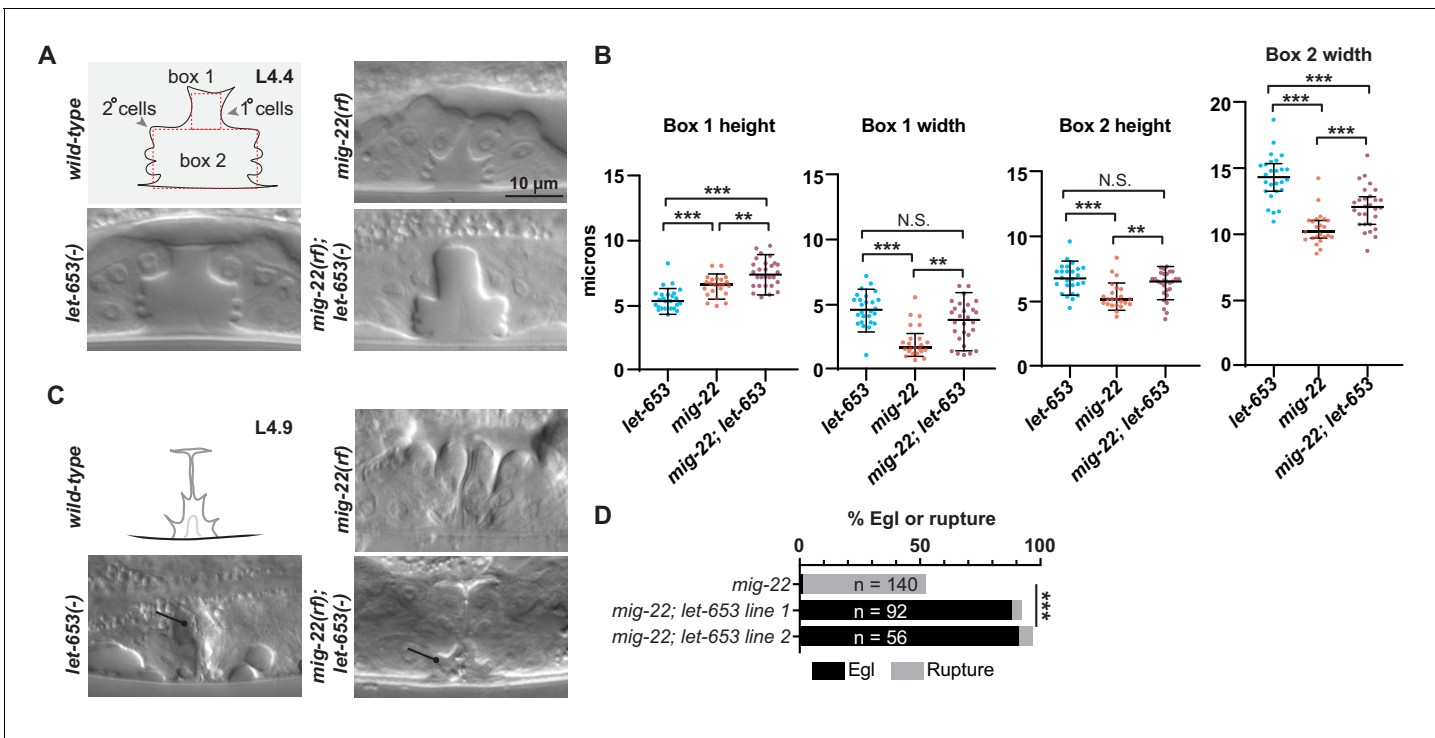

**Figure 11.** Chondroitin and LET-653 have both lumen-expanding and lumen-constraining roles. (A) Loss of *let-653* suppressed the *mig-22(rf)* Sqv phenotype and caused over-inflation of the dorsal lumen (n = 28). Alleles used were *let-653(cs178)* and *mig-22(k141rf)*. (B) Lumen dimensions at the L4.4 stage were quantified as indicated in panel A. *let-653* single mutant dimensions appeared similar to *wild-type* at this stage (n = 26; see *Figure 8— figure supplement 1*). p values derived from Mann–Whitney U test; **p<0.01, ***p<0.0001. Box 1 height, *mig-22* vs. *mig-22; let-653* p=0.002. Box 1 width, *mig-22* vs. *mig-22; let-653* p=0.001. Box 2 height, *mig-22* vs. *mig-22; let-653* p=0.003. (C) At late L4 stages, some *let-653* single mutants (n = 3/16) and *mig-22(rf); let-653* double mutants (n = 5/22) had disorganized material within the vulva lumen (line). (D) Two independently generated *mig-22(rf); let-653* double mutant strains were analyzed, and both were severely egg-laying defective. ***p<0.0001, Fisher's exact test.

The online version of this article includes the following source data for figure 11:

**Source data 1.** Measurements of *WT* and mutant L4.4 stage vulvas.

uniform hydrostatic expansion force. Rather, as shown here, chondroitin proteoglycans act within a complex luminal scaffold that likely exerts, resists, and distributes multiple different types of vulva cell- and lumen-shaping forces.

## Discussion

The diameter of a tube lumen is ultimately determined by the shape and organization of the cells that surround that lumen. Two well-known determinants of cell shape are the cytoskeleton and the ECM. Here, we showed that the luminal aECM within the developing *C. elegans* vulva tube has a structural and functional complexity that rivals that of the cytoskeleton. The cytoskeleton consists of multiple dynamic and interacting components (actin, microtubules, and intermediate filaments) that are organized into both cytosolic and membrane-anchored fibrils and webs (*Fletcher and Mullins, 2010*). These cytoskeletal elements can exert both pushing and pulling forces on cell membranes (*Fletcher and Mullins, 2010*). Similarly, the vulva luminal matrix contains a variety of both free and seemingly membrane-attached structural elements. These elements are cell-type specific and highly dynamic over the course of tube morphogenesis. Removal of individual aECM elements, or sets of elements, has distinct effects on cell and lumen shape, revealing both lumen- expanding and lumen constricting roles. Ultimately, these data reveal a complex and dynamic aECM which offers a powerful model for investigating aECM assembly, remodeling, and tube-shaping capacity.

### Multiple types of matrix shape the vulva lumen

Although it has been clear for over a decade that chondroitin GAGs are required to inflate the vulva lumen (*Herman et al., 1999*; *Hwang and Horvitz, 2002a*; *Hwang and Horvitz, 2002b*; *Hwang et al., 2003a*; *Hwang et al., 2003b*), other aECM factors involved in shaping had not been previously described. Here, we showed that the vulva aECM contains multiple discrete elements (*Figure 12*). First, a granular matrix containing the CPG FBN-1 (and likely many others) fills the luminal cavity by mid-L4. Second, distinct types of membrane-proximal aECMs line different vulva cell types at different stages; these aECM layers contain ZP domain, eLRRon and lipocalin proteins and appear to be analogous to the embryonic sheath that lines the epidermis (*Mancuso et al., 2012*; *Priess and Hirsh, 1986*; *Vuong-Brender et al., 2017*). Third, a stalk-like core structure forms within the central lumen; this core is marked by the ZP protein LET-653 via its PAN domains (*Gill et al., 2016*). Finally, the core attaches to different aECM-covered cell surfaces via ventro-lateral fibrils, also marked by LET-653 PAN domains. Each of these elements may play different roles in lumen shaping (*Figure 12*).

   The six matrix proteins analyzed here are probably a very small subset of the total components that make up the various vulva aECM structures, and for the most part their roles in building these structures remain to be tested. We showed that *let-653* is broadly important for proper morphology of the membrane-proximal matrices over the AC/utse, vulE/F, and vulC/D, and for restricting fibrillar structures to their proper ventral locations. Neither LET-653 nor any of the other aECM proteins

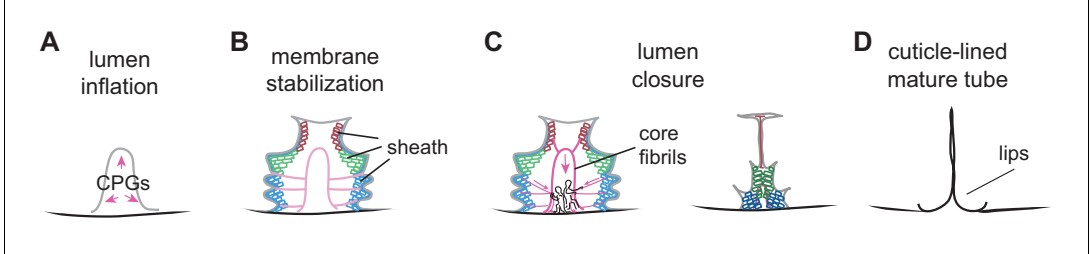

**Figure 12.** Model for aECM-dependent shaping of the vulva lumen during morphogenesis. (**A**) The vulva lumen is initially expanded by chondroitin proteoglycans (pink arrows). Black; cuticle. Gray; apical membrane. (**B**) A membrane-proximal aECM appears alongside matrix fibrils and a central core to halt and/or stabilize vulva expansion. vulE/F aECM; red. vulC/D aECM; green. vulA/B aECM; blue. (**C**) The lumen narrows in the anterior-posterior axis. We propose that the central core and fibrils attach to the aECM and underlying membranes and pull ventrally, anteriorly, and posteriorly to shape the vulva lumen. The aECM changes over time; transient components turn over and the membrane-proximal matrix develops cuticle-like features. (**D**) By adulthood, the lumen narrows into a slit and is lined by cuticle.

tested is required to build the luminal core structure, but loss of this structure in *lin-12*/Notch mutants suggests that both 1°- and 2°-derived factors are involved.

## Apical matrix may generate, counteract, and distribute multiple forces during lumen shaping

Prior work showed that the vulva is expanded by a combination of pushing forces by chondroitin GAGs (*Hwang et al., 2003a*) and actin-myosin constriction by 2° cells, which may then focus the GAG-dependent forces dorsally (*Yang et al., 2017a*). The data here suggest that GAGs not only inflate the lumen, but also play other roles. First, chondroitin depletion does not affect all vulva cells in the same way; in *mig-22(rf)* mutants, some cells have expanded apical domains, while others have shortened apical domains. Second, in the context of *let-653* loss, *mig-22(rf)* actually causes increased dorsal lumen expansion, revealing a lumen-constraining role for both LET-653 and chondroitin (*Figure 11*). Finally, the predominant defect in *fbn-1* single mutants is not in lumen inflation, but instead in vulva eversion, showing that GAG-modified proteins have diverse functions in vulva shaping.

The other aECM factors described here may counteract and distribute the forces exerted by GAGs and the cytoskeleton, or may themselves generate additional types of forces. The vulE/F aECM and central core component LET-653 restrains CPG-dependent lumen inflation in early-to-mid-L4 stages, but also helps maintain lumen inflation during the subsequent steps of vulva morphogenesis. Interestingly, LET-653 impacts these later steps even though the functional LET-653(ZP) domain is no longer visible at those stages. LET-653(ZP) and other aECM components may form a membrane-anchored scaffold whose assembly at mid-L4 initially 'locks in' a particular lumen size to prevent further CPG-dependent inflation (*Figure 12*). This transient scaffold could then serve as a template for recruitment and assembly of later matrix factors, including cuticle collagens, that will sculpt the final structure. Through unknown mechanical connections with membranes and the cytoskeleton, aECM factors also may influence the cytoskeletal organization of vulva cells to promote their rearrangements and cell shape changes. Connections among aECM, vulva cells, and sex muscles could serve as anchor points to transmit muscle-generated forces, as proposed for epidermal elongation in the embryo (*Vuong-Brender et al., 2017*). Finally, the LET-653-marked core structure and associated fibrils appear to constrict during lumen narrowing, and such matrix reorganization could potentially exert a pulling force on apical membranes (*Figure 12*), as has been proposed for alae (cuticle ridge) formation (*Sapio et al., 2005*).

Multiple aECM factors likely cooperate in vulva lumen shaping with LET-653. Most aECM single mutants, including *let-653,* have only subtle vulva shaping defects and a low percentage of egg-laying defective adults, despite drastic defects in matrix organization (*Figures 8* and *9*). This is in contrast to the much more penetrant and dramatic phenotypes observed in these same mutants in shaping narrow excretory system tubes (*Gill et al., 2016*; *Mancuso et al., 2012*) or the embryonic epidermis (*Vuong-Brender et al., 2017*). The vulva appears to be less sensitive to defects in aECM than these other tissues, possibly because its large lumen can tolerate many irregularities in cell shape and still remain passable for egg-laying.

## LET-653 and cell-type-specific partners may traffic through large secretory vesicles in vulF

Individual aECM components appear and disappear from individual cell surfaces at specific timepoints. These localization patterns suggest careful regulation. How are these proteins, and their corresponding aECM layers, built and broken down at the correct times and places? Part of the answer may lie in time- and cell-type-specific secretion.

Many secreted matrix proteins are packaged and trafficked via specialized vesicles. For example, collagens are secreted from extra-large vesicles (*Malhotra and Erlmann, 2015*) and then processed after secretion to enable their assembly (*Holmes et al., 2018*). Different ZP proteins secreted from the same cell are sorted into separate pools of vesicles, possibly to prevent their premature association (*Jovine et al., 2007*). In fact, we observed large vesicles emptying material into the vulva lumen from vulF (*Figure 4*). LET-653 is a potential cargo within these secretory vesicles and may traffic with one or more partners that form the membrane-proximal aECM over vulE/F. In the absence of LET-653, vesicle contents appear to aggregate abnormally as soon as they are exposed to the luminal environment. Although LET-653 is expressed by all vulva cell types, it may traffic in different types of

vesicles or with different partners in each cell type, explaining why LET-653(ZP) does not form the same type of matrix over all vulva cells, and why abnormal vesicles were not observed in other vulva cells in *let-653* mutants.

The vulF secretory vesicles resemble mucin vesicles found in goblet cells of the mammalian lung and gut. Tightly compressed mucin packets in those vesicles are thought to expand rapidly upon reaching the luminal environment due to changes in pH and salt conditions *Birchenough et al., 2015*; the CFTR ion channel, which is mutated in cystic fibrosis patients, is expressed on adjacent cell types (*Kreda et al., 2012*) and is important for establishing the proper luminal environment for mucin expansion to occur (*Kreda et al., 2012*). Given that vulF secretory vesicles empty into narrow lumen compartments between vulF and vulE, it is possible that vulE expresses key ion channels or other matrix factors that are important for proper assembly of the extruded matrix.

## The vulva as a system for visualizing and dissecting aECM trafficking and assembly

Classic TEM studies in many systems have shown that aECMs are layered structures (*Chappell et al., 2009*; *Johansson et al., 2011*). However, a major challenge for understanding aECMs has been the difficulty in visualizing them in a more high-throughput manner. The large size of the vulva lumen, combined with the transparency of *C. elegans*, allowed us to visualize the various aECM elements by light microscopy in a way that is unprecedented in other systems. The vulva therefore provides a very powerful system for addressing further mechanistic questions regarding how aECM components traffic to the apical surface, how aECM structures are assembled and remodeled, and how they ultimately impact cell and tube shape.

# Materials and methods

## Worm strains, alleles and transgenes

See *Table 1* and Key Resources Table for a complete list of all strains, alleles and transgenes used. All strains were derived from Bristol N2 and were grown at 20 °C under standard conditions (*Brenner, 1974*). *let-653* and *let-4* mutants were obtained from mothers rescued with wild-type transgenes expressed in the excretory system under the control of the *lin-48* or *grl-2* promoters (*Forman-Rubinsky et al., 2017*). Prior tissue-specific rescue experiments showed that LET-653 (*Gill et al., 2016*) and LET-4 (*Mancuso et al., 2012*) have little or no ability to travel between tissues. Consistent with this, *lin-48pro*::LET-653::SfGFP did not drive any detectable GFP expression in the vulva. *sqv-5* mutants were obtained from heterozygous mothers. All other mutants were obtained from homozygous mothers.

Transgene *aaaIs12 [fbn-1pro::FBN-1::mCherry; ttx-3pro::GFP]* was derived from array *aaaEx78* (*Katz et al., 2018*) and expresses full-length FBN-1 tagged internally with mCherry inserted just prior to the ZP domain. *aaaEx78* rescued *fbn-1(tm290)* molt defects and other lethality from 85% (n = 236) in non-transgenic siblings to 10% (n = 1676) in transgenic animals.

## CRISPR/Cas9-mediated generation of reporters

To generate LET-653::SfGFP and SfGFP::LPR-3 fusions via CRISPR, a self-excising cassette (SEC) vector containing SfGFP was generated (pJC39). SfGFP was amplified and inserted a larger fragment of the SEC vector pDD282 (*Dickinson et al., 2013*) via PCR sewing. The large fragment was inserted into pDD282 as a NaeI – BglII fragment. Henceforth, CRISPR was carried out as described (*Dickinson et al., 2015*). Briefly, LET-653 or LPR-3 homology arms were inserted into the resulting plasmid and the relevant PAM sites were then mutated via site-directed mutagenesis. sgRNAs were generated via site-directed mutagenesis that inserted a primer encoding the gRNA before a U6 promoter in the plasmid pDD162 (*Dickinson et al., 2013*). These plasmids were injected into N2 hermaphrodites. F2 progeny were screened by microscopy for Hygromycin resistance and/or SfGFP fluorescence. Insertions were verified by PCR and Sanger sequencing. SfGFP was inserted immediately before the LET-653 stop codon or immediately following the LPR-3 signal peptide. An SEC inserted into an intron within the SfGFP coding sequence was removed via heat shock (*Dickinson et al., 2015*). Excision was confirmed by PCR and Sanger sequencing.

**Table 1.** Strains used in this work.

| Strain | Genotype |
| --- | --- |
| ARF335 | fbn-1(tm290) III; aaaEx78 [fbn-1pro::FBN-1::mCherry; ttx-3pro::GFP] (**Katz et al., 2018**) |
| ARF379 | aaaIs12 [fbn-1pro::FBN-1::mCherry; ttx-3pro::GFP] V (**Katz et al., 2018**) |
| ARF359 | upIs1 [MUP-4::GFP; rol-6(su1006)] V; aaaEx78 [fbn-1pro::FBN-1::mCherry; ttx-3pro::GFP] (**Hong et al., 2001**; **Katz et al., 2018**) |
| HM24 | upIs1 V; aaaIs12 V (**Hong et al., 2001**; **Katz et al., 2018**) |
| GOU2043 | vab-10(cas602 [vab-10a::gfp]) I (**Yang et al., 2017b**) |
| JU486 | mfIs4 [egl-17pro::YFP; daf-6pro::CFP; unc-119(+)] (**Mok et al., 2015**) |
| ML2482 | noah-1(mc68 [NOAH-1::mCH(int)]) I (**Vuong-Brender et al., 2017**) |
| ML2547 | sym-1(mc85 [SYM-1::GFP]) X (**Vuong-Brender et al., 2017**) |
| ML2615 | dlg-1(mc103 [DLG-1::GFP]) X (**Vuong-Brender et al., 2017**) |
| N2 | WT |
| NF68 | mig-22(k141) III (**Suzuki et al., 2006**) |
| SP2163 | sym-1(mn601) X (**Niwa et al., 2009**) |
| UP2386 | csEx358 [lpr-1pro::LET-653b; unc-119pro::GFP] (**Gill et al., 2016**) |
| UP3244 | let-653(cs178) IV; csEx358 (**Gill et al., 2016**) |
| UP3342 | let-653(cs178) IV; csEx766 [lin-48pro::LET-653b::SfGFP; myo-2pro::GFP] (**Forman-Rubinsky et al., 2017**) |
| UP3349 | aaaIs12 V; csIs64 [let-653pro::LET-653b::SfGFP; rol-6(su1006)] |
| UP3353 | let-653(cs178) IV; aaaIs12 V; csEx766 |
| UP3422 | csIs66 [let-653pro::LET-653(ZP)::SfGFP; let-653pro::PH::mCherry] X |
| UP3444 | sqv-5(n3611)/hT2 [bli-4(e937) let-?(q782) qIs48] I,III; csIs66 X |
| UP3462 | let-653(cs178) IV; csIs66 X |
| UP3666 | lpr-3(cs250 [ssSfGFP::LPR-3]) X |
| UP3693 | noah-1(mc68 [NOAH-1::mCH(int)]) I; lpr-3(cs250 [ssSfGFP::LPR-3]) X |
| UP3746 | let-653(cs262 [LET-653::SfGFP]) IV |
| UP3756 | let-4(cs265 [ssmCherry::LET-4]) X |
| UP3757 | dpy-19(e1259) lin-12(n137)/hT2 [bli-4(e937) let-?(q782) qIs48] I,III; csIs66 X |
| UP3758 | unc-32(e189) lin-12(n137 n720)/hT2 [bli-4(e937) let-?(q782) qIs48] I,III; csIs66 X |
| UP3788 | noah-1(mc68 [NOAH-1::mCH(int)]) I; let-653(cs262 [LET-653::SfGFP]) IV |
| UP3856 | let-653(cs262 [LET-653::SfGFP]) IV; let-4(cs265 [ssmCherry::LET-4]) X |
| UP3861 | muIs27 [MIG-2::GFP; dpy-20+]; let-4(cs265 [ssmCherry::LET-4]) X (**Honigberg and Kenyon, 2000**) |
| UP3939 | let-4(mn105) X; csEx819 [grl-2pro::LET-4; myo-2pro::mRFP] (**Forman-Rubinsky et al., 2017**) |
| UP3967 | mig-22(k141) III; let-653(cs178) IV; csEx766 |
| UP3968 | mig-22(k141) III; let-653(cs178) IV; csEx766 |
| UP3970 | mig-22(k141) III; csIs66 X |
| UP3979 | let-653(cs178) IV; dlg-1(mc103 [DLG-1::GFP]) X; csEx766 |
| UP3995 | muIs28 [MIG-2::GFP; unc-31+] (**Honigberg and Kenyon, 2000**) |
| UP3966 | noah-1(mc68 [NOAH-1::mCH(int)]) I; let-653(cs178) IV; lpr-3(cs250 [ssSfGFP::LPR-3]) X; csEx766 |
| UP4004 | mig-22(k141) III; muIs28 |
| UP4005 | let-653(cs178) IV; muIs28; csEx766 |
| UP4014 | noah-1(mc68 [NOAH-1::mCH(int)]) I; mig-22(k141) III; lpr-3(cs250 [ssSfGFP::LPR-3]) X |
| UP4025 | noah-1(mc68 [NOAH-1::mCH(int)])/hT2 [bli-4(e937) let-?(q782) qIs48] I; unc-32(e189) lin-12(n137 n720)/hT2 [bli-4(e937) let-?(q782) qIs48] III |
| UP4027 | mig-22(k141) III; let-653(cs262 [LET-653::SfGFP]) IV |
| UP4038 | unc-32(e189) lin-12(n137 n720)/hT2 [bli-4(e937) let-?(q782) qIs48] I,III; lpr-3(cs250 [ssSfGFP::LPR-3]) X |

*Table 1 continued on next page*

*Table 1 continued*

| Strain | Genotype |
|--------|----------|
| UP4039 | *mig-22(k141) III; mfIs4* |
| UP4040 | *let-653(cs178) IV; mfIs4; csEx766* |
| UP4042 | *unc-32(e189) lin-12(n137 n720)/hT2 [bli-4(e937) let-?(q782) qIs48] I,III; let-653(cs262 [LET-653::SfGFP]) IV* |
| UP4043 | *dpy-19(e1259) lin-12(n137)/hT2 [bli-4(e937) let-?(q782) qIs48] I,III; lpr-3(cs250 [ssSfGFP::LPR-3]) X* |
| UP4044 | *dpy-19(e1259) lin-12(n137)/hT2 [bli-4(e937) let-?(q782) qIs48] I,III; let-653(cs262 [LET-653::SfGFP]) IV* |
| UP4045 | *noah-1(mc68 [NOAH-1::mCH(int)]) I; dpy-19(e1259) lin-12(n137) III* |
| UP4047 | *noah-1(mc68 [NOAH-1::mCH(int)]) I; muIs28* |

The mCherry::LET-4 fusion was generated using the SapTrap method (*Schwartz and Jorgensen, 2016*). LET-4 homology arms were inserted into plasmid pMLS291 using SapI digestion and ligation. The resulting plasmid was co-injected into N2 hermaphrodites with a plasmid containing relevant sgRNAs inserted into pDD162 (*Dickinson et al., 2013*). F2 progeny were screened by microscopy for mCherry fluorescence. Insertions were verified by PCR and Sanger sequencing. mCherry was inserted immediately after the LET-4 signal sequence.

All CRISPR fusion strains were evaluated for viability. *let-653(cs262[LET-653::SfGFP])* was 93% viable (n = 195), *lpr-3(cs250[SfGFP::LPR-3])* was 98% viable (n = 137), and *let-4(cs265[mCherry::LET-4])* was 100% viable (n = 154).

## Staging and microscopy

Larvae were staged by vulva morphology. Fluorescent, brightfield, differential interference contrast (DIC), and Dodt (an imaging technique that simulates DIC) (*Dodt and Zieglgänsberger, 1990*) images were captured on a compound Zeiss Axioskop fitted with a Leica DFC360 FX camera or with a Leica TCS SP8 confocal microscope (Leica, Wetzlar Germany). Images were processed and merged using ImageJ.

For TEM, L4 hermaphrodites from N2 (wild-type) or UP3342 (*let-653(cs178); csEx766[lin-48pro:: LET-653::SfGFP; myo-2pro::GFP]*) strains were fixed by high pressure freezing followed by freeze substitution into osmium tetroxide in acetone (*Weimer, 2006*), and then rinsed and embedded into LX112 resin and cut into serial thin sections of approximately 70 nm each. Sections were observed on a JEM-1010 transmission electron microscope (Jeol, Peabody Massachusetts). Images were processed in ImageJ and manually pseudo-colored in Adobe Illustrator (Adobe, San Jose California). We imaged vulvas for n = 1 mid-L4 N2, n = 2 late L4 N2, and n = 1 mid-L4 UP3342. Data for all animals are shown.

## Lumen measurements

Vulva dimensions were measured using the box tool in ImageJ. Boxes were drawn within the lumen in DIC images and their length and width were recorded. A minimum of 10 animals were examined for each experiment. vulD aspect ratio was measured by tracing the cell outline from a single sagittal confocal Z-slice and then using the shape measurement tool in ImageJ. The two vulD cells from each animal were treated separately. All measurements were performed by a researcher blinded to genotype. Statistics were calculated using Prism software (Graphpad, San Diego California) using two-tailed Mann–Whitney U tests.

## Acknowledgements

We thank Ken Nguyen (Albert Einstein School of Medicine) and Biao Zuo and Inna Martynyuk (UPenn Electron Microscopy Resource Lab) for assistance with TEM, Andrea Stout (UPenn CDB Microscopy Core) for training and assistance with confocal imaging, Lily Zekavat for assistance with *lin-12* experiments, Michel Labouesse for strains, and John Murray, Nick Serra, Susanna Birnbaum, Hasreet Gill, Dave Matus, and members of the Matus laboratory for helpful discussions and comments on the manuscript. Some strains were provided by the CGC, which is funded by the NIH Office of Research

Infrastructure Programs (P40 OD010440). This work was funded by NIH grants R01GM58540, R01GM125959, and R35GM136315 to M.V.S., T32 GM008216 and T32 AR007465 to J.D.C., and NIH OD010943 to D.H.H, and by ACS grant RSG-12-149-01-DDC to A.R.F.

## Additional information

### Funding

| Funder | Grant reference number | Author |
|---|---|---|
| National Institute of General Medical Sciences | R01GM58540 | Meera V Sundaram |
| American Cancer Society | RSG-12-149-01-DDC | Alison R Frand |
| National Institute of General Medical Sciences | R01GM125959 | Meera V Sundaram |
| National Institute of General Medical Sciences | T32 GM008216 | Jennifer D Cohen |
| National Institute of Arthritis and Musculoskeletal and Skin Diseases | T32 AR007465 | Jennifer D Cohen |
| Office of the Director | OD010943 | David H Hall |
| National Institute of General Medical Sciences | R35GM136315 | Meera V Sundaram |

The funders had no role in study design, data collection and interpretation, or the decision to submit the work for publication.

### Author contributions

Jennifer D Cohen, Conceptualization, Formal analysis, Investigation, Visualization, Writing - original draft; Alessandro P Sparacio, Alexandra C Belfi, Investigation, Visualization, Writing - review and editing; Rachel Forman-Rubinsky, Hannah Maul-Newby, Investigation, Writing - review and editing; David H Hall, Resources, Funding acquisition, Investigation, Project administration, Writing - review and editing; Alison R Frand, Conceptualization, Supervision, Funding acquisition, Project administration, Writing - review and editing; Meera V Sundaram, Conceptualization, Supervision, Funding acquisition, Investigation, Visualization, Writing - original draft, Project administration

### Author ORCIDs

David H Hall http://orcid.org/0000-0001-8459-9820
Hannah Maul-Newby http://orcid.org/0000-0003-2187-7891
Alison R Frand http://orcid.org/0000-0001-5972-989X
Meera V Sundaram https://orcid.org/0000-0002-2940-8750

### Decision letter and Author response

Decision letter https://doi.org/10.7554/eLife.57874.sa1
Author response https://doi.org/10.7554/eLife.57874.sa2

## Additional files

### Supplementary files

• Transparent reporting form

### Data availability

All data generated or analysed during this study are included in the manuscript and supporting files. Source data files have been provided for graphs in Figures 8, 8-1,10, 11.

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

# Appendix 1

**Appendix 1—key resources table**

| Reagent type (species) or resource | Designation | Source or reference | Identifiers | Additional information |
|---|---|---|---|---|
| Gene (*Caenorhabditis elegans*) | *fbn-1 III* | Wormbase WS277 | *fbn-1*; ZK783.1 | FiBrilliN homolog |
| Gene (*C. elegans*) | *let-653 IV* | Wormbase WS277 | *let-653*; C29E6.1 | LEThal; ZP domain |
| Gene (*C. elegans*) | *let-4 X* | Wormbase WS277 | *let-4*; C44H4.2 | LEThal; eLRRon |
| Gene (*C. elegans*) | *lin-12 III* | Wormbase WS277 | *lin-12*; R107.8 | abnormal cell LINeage; Notch-related |
| Gene (*C. elegans*) | *mig-22 III* | Wormbase WS277 | *mig-22*; PAR2.4 | abnormal cell MIGration; chondroitin polymerizing factor |
| Gene (*C. elegans*) | *mup-4 III* | Wormbase WS277 | *mup-4*; K07D8.1 | MUscle Positioning; matrilin-related |
| Gene (*C. elegans*) | *noah-1 I* | Wormbase WS277 | *noah-1*; C34G6.6 | NOmpA homolog; ZP domain |
| Gene (*C. elegans*) | *sqv-5 I* | Wormbase WS277 | *sqv-5*; T24D1.1 | SQuashed Vulva; chondroitin sulfate synthase |
| Gene (*C. elegans*) | *sym-1 X* | Wormbase WS277 | *sym-1*; C44H4.3 | SYnthetic lethal with Mec; eLRRon |
| Gene (*C. elegans*) | *vab-10 I* | Wormbase WS277 | *vab-10*; ZK1151.1 | VAriaBle abnormal morphology; dystonin ortholog |
| Genetic reagent (*C. elegans*) | *fbn-1 (tm290) III* | *Kelley et al., 2015* | strain ARF335 | 604 bp deletion/frameshift; presumed null |
| Genetic reagent (*C. elegans*) | *let-4 (mn105) X* | *Mancuso et al., 2012* | strain UP3939 | *Q305ochre*; presumed null |
| Genetic reagent (*C. elegans*) | *let-4(cs265 [mCherry:: LET-4]) X* | This study | strain UP3756 | mCherry fused to endogenous LET-4 near its N-terminus by Cas9-triggered homologous recombination |
| Genetic reagent (*C. elegans*) | *let-653 (cs178) IV* | *Gill et al., 2016* | strain UP3244 | *C54ochre*; presumed null |
| Genetic reagent (*C. elegans*) | *let-653(cs262 [LET-653:: SfGFP]) IV* | This study | strain UP3746 | SfGFP fused to endogenous LET-653 at its C-terminus by Cas9-triggered homologous recombination |

*Continued on next page*

*Appendix 1—key resources table continued*

| Reagent type (species) or resource | Designation | Source or reference | Identifiers | Additional information |
|---|---|---|---|---|
| Genetic reagent (*C. elegans*) | *lin-12 (n137n720) III* | (*Sternberg and Horvitz, 1989*). *Caenorhabditis* Genetics Center (CGC). | strain MT2343 | presumed null |
| Genetic reagent (*C. elegans*) | *lin-12 (n137) III* | (*Greenwald et al., 1983*). CGC. | strain MT2343 | *S872F;* hypermorph |
| Genetic reagent (*C. elegans*) | *lpr-3(cs250 [SfGFP:: LPR-3]) X* | This study | strain UP3666 | SfGFP fused to endogenous LPR-3 near its N-terminus by Cas9-triggered homologous recombination |
| Genetic reagent (*C. elegans*) | *mig-22(k141) III* | (*Suzuki et al., 2006*). CGC. | strain NF68 | *G227E;* hypomorph |
| Genetic reagent (*C. elegans*) | *noah-1(mc68 [NOAH-1:: mCherry]) I* | (*Vuong-Brender et al., 2017*). Michel Labouesse lab. | strain ML2482 | mCherry fused to endogenous NOAH-1 at internal site by Cas9-triggered homologous recombination |
| Genetic reagent (*C. elegans*) | *sqv-5(n3611) I* | (*Hwang et al., 2003b*). CGC. | strain MT10996 | deletion; presumed null |
| Genetic reagent (*C. elegans*) | *sym-1 (mn601) X* | (*Davies et al., 1999*). CGC. | strain SP2163 | *Q275ochre;* presumed null |
| Genetic reagent (*C. elegans*) | *sym-1(mc85 [SYM-1:: GFP]) X* | (*Vuong-Brender et al., 2017*). Michel Labouesse lab. | strain ML2547 | GFP fused to endogenous SYM-1 at its C-terminus by Cas9-triggered homologous recombination |
| Genetic reagent (*C. elegans*) | *vab-10 (cas602 [VAB-10a:: GFP])* | (*Yang et al., 2017b*). CGC. | strain GOU2043 | GFP fused to endogenous VAB-10a at its C-terminus by Cas9-triggered homologous recombination |
| Genetic reagent - Transgene (*C. elegans*) | *aaaIs12 [fbn-1pro::FBN-1:: mCherry; ttx-3pro::GFP]* | This study | *aaaIs12,* strain ARF379 | Transgene expressing full-length FBN-1 tagged internally with mCherry inserted just prior to the ZP domain |
| Genetic reagent - Transgene (*C. elegans*) | *csEx766 [lin-48pro:: LET-653:: SfGFP; myo-2pro:: GFP]* | *Forman-Rubinsky et al., 2017* | *csEx766* strain UP3342 | Duct-specific rescue transgene |
| Genetic reagent - Transgene (*C. elegans*) | *csEx819 [grl-2pro:: LET-4; myo-2p:: mRFP]* | *Forman-Rubinsky et al., 2017* | *csEx819* strain UP3939 | Duct/pore-specific rescue transgene |
| Genetic reagent - Transgene (*C. elegans*) | *csIs64 [let-653pro:: LET-653:: SfGFP; lin-48pro:: mRFP]* | *Gill et al., 2016* | *csIs64* strain UP3070 | Transgene expressing SfGFP-tagged LET-653 |

*Continued on next page*

*Appendix 1—key resources table continued*

| Reagent type (species) or resource | Designation | Source or reference | Identifiers | Additional information |
|---|---|---|---|---|
| Genetic reagent - Transgene (*C. elegans*) | csIs66 [let-653pro:: SfGFP:: LET-653(ZP); let-653pro:: PH:: mCherry; lin-48pro:: mRFP] | *Cohen et al., 2019* | csIs66 strain UP3422 | Transgene expressing SfGFP-tagged LET-653(ZP) domain |
| Genetic reagent - Transgene (*C. elegans*) | mfIs4[egl-17pro::YFP; daf-6pro::CFP; unc-119+] | (*Félix, 2007*). CGC. | mfIs4 strain JU486 | Transgene expressing CFP in primary vulva descendants and YFP in secondary vulva descendants. |
| Genetic reagent - Transgene (*C. elegans*) | muIs28 [mig-2pro:: MIG-2::GFP; unc-31+] | (*Honigberg and Kenyon, 2000*). CGC. | muIs28 strain CF693 | Transgene expressing GFP-tagged MIG-2 (membrane marker) |
| Genetic reagent - Transgene (*C. elegans*) | upIs1 [mup-4::GFP + rol-6(su1006)] | (*Hong et al., 2001*). CGC. | upIs1; strain EE86 | Transgene expressing GFP-tagged MUP-4 |
| recombinant DNA reagent | pDD282 (plasmid) | *Dickinson et al., 2013* | Addgene plasmid #66823 | GFPSEC3xFlag vector with ccdB markers for cloning homology arms |
| recombinant DNA reagent | pDD162 (plasmid) | *Dickinson et al., 2013* | Addgene plasmid #47549 | eft-3p::Cas9 + empty sgRNA plasmid |
| recombinant DNA reagent | pJC39 | This study | | pDD282 with GFP replaced by SfGFP |
| recombinant DNA reagent | pJC49 | This study | | *let-653* homology repair template generated by PCR with OJC201+ OJC202 and oJC203+ oJC204 and cloned into pJC39 |
| recombinant DNA reagent | pJC50 | This study | | *let-653* sgRNA sequence (5'-TTGAGATATTACG TTCGAAC-3') cloned into pDD162 |
| recombinant DNA reagent | pJC67 | This study | | *let-4* homology repair template generated by PCR with oJC269+oJC270 and oJC271+oJC282 and cloned into pMLS291 |
| recombinant DNA reagent | pJC68 | This study | | *let-4* sgRNA sequence (5'- GACTCCAGGA CAAGCATTTG −3') cloned into pDD162 |
| recombinant DNA reagent | pMLS291 (plasmid) | *Schwartz and Jorgensen, 2016* | Addgene plasmid #73724 | SapTrap vector with mCherry |
| recombinant DNA reagent | pMLS328 (plasmid) | *Schwartz and Jorgensen, 2016* | Addgene plasmid #73717 | eft-3p::2xNLS-Cre, unc-119+, for SEC excision |
| recombinant DNA reagent | pRFR60 | This study | | *lpr-3* sgRNA sequence (5'-TTTGGCTACGA CGTTAGCTG −3') cloned into pDD162 |

*Continued on next page*

*Appendix 1—key resources table continued*

| Reagent type (species) or resource | Designation | Source or reference | Identifiers | Additional information |
|---|---|---|---|---|
| recombinant DNA reagent | pRFR70 | This study | | *lpr-3* homology repair template generated by PCR with oRFR69+oRFR90 and oRFR71+oRFR72 and cloned into pJC39 |
| sequence-based reagent | oJC201 | This study | PCR primer | 5'_ACGTTGTAAA ACGACGGCCAGTC GCCGGCA-CAAAAA TCAGTCTATCATTCC_3' |
| sequence-based reagent | oJC202 | This study | PCR primer | 5'_TCCAGTGAAAAG TTCTTCTCCTTTGC TGAT-GATGTTTgC AGTTCGAACG_3' |
| sequence-based reagent | oJC203 | This study | PCR primer | 5'_CGTGATTACAAG GATGACGATGACA AGAGA-TGAAAATA CACACAAAAAATG_3' |
| sequence-based reagent | oJC204 | This study | PCR primer | 5'_TCACACAGGAAA CAGCTATGACCATGT TAT-GTCTGGTAGCT GCTTTGATGATGG_3' |
| sequence-based reagent | oJC269 | This study | PCR primer | 5'_gtgGCTCTTCgTG Ggtttaaacacgtattt cacacatttttcag_3' |
| sequence-based reagent | oJC270 | This study | PCR primer | 5'_gtgGCTCTTCg CATTCCAGGACAA GCATTTGTCGAG_3' |
| sequence-based reagent | oJC271 | This study | PCR primer | 5'_gtgGCTCTTCg GGTGTCATTACTC AAGCGTGCTTC_3' |
| sequence-based reagent | oJC282 | This study | PCR primer | 5'_GTGGCTCTTCg TACGATGGCACTGCA GTCATATTG_3' |
| sequence-based reagent | oRFR69 | This study | PCR primer | 5'_ACGTTGTAAAACG ACGGCCAGTCGCCGGC Acatataataaagca ttttgtctg_3' |
| sequence-based reagent | oRFR90 | This study | PCR primer | 5'_TCCAGTGAAAAGTT CTTCTCCTTTGCTgATGC CTAGTGCgACAGCTAAC_3' |
| sequence-based reagent | oRFR71 | This study | PCR primer | 5'_CGTGATTACAAGGAT GACGATGACAAGAGAG CTATTAGCGAAGCAGAC GTACC_3' |
| sequence-based reagent | oRFR72 | This study | PCR primer | 5'_TCACACAGGAAAC AGCTATGACCATGTTA TCGGTAACGGTCTT GACTCCGGC_3' |
| software, algorithm | ImageJ | NIH | | |
| software, algorithm | Adobe Illustrator | Adobe | | |
| software, algorithm | Imaris | Bitplane | | |
| software, algorithm | Prism | Graphpad | | |

