## [Decision Letter]

**Acceptance summary:**

Understanding how cellular tubes are molded and protected by apical extracellular matrices that line their lumens is an extremely interesting question and an exciting research direction.

**Decision letter after peer review:**

Thank you for submitting your article "A multi-layered and dynamic apical extracellular matrix shapes the vulva lumen in *Caenorhabditis elegans*" for consideration by *eLife*. Your article has been reviewed by three peer reviewers, and the evaluation has been overseen by a Reviewing Editor and Piali Sengupta as the Senior Editor. The following individual involved in review of your submission has agreed to reveal their identity: David R Sherwood (Reviewer #1).

The reviewers have discussed the reviews with one another and the Reviewing Editor has drafted this decision to help you prepare a revised submission.

All three reviewers thought the paper made important progress in our understanding of ECM and organogenesis. All three reviewers showed strong enthusiasm towards this manuscript. Their main suggestions for revision is to characterize the loss-of-function mutants or RNAi of VAB-10a, MUP-4 and MUA-3. We look forward to reading the revisions.

Summary:

In this work, "A multi-layered and dynamic apical extracellular matrix shapes the vulva lumen in *Caenorhabditis elegans*", the authors use functional fluorescently tagged apical matrix components, confocal microscopy, transmission electron microscopy, and genetics to characterize components and function of apical extracellular matrix (aECM) lining the *C. elegans* vulval lumen. The authors show that the aECM has distinct elements, including a granular matrix that fills the entire vulval lumen, and central matrix fibrils that form a stalk-like core. The matrix is highly dynamic-its composition varies depending on the developmental stage and the identity of the vulval cells it contacts. Finally, genetic perturbation of vulval matrix components suggest that the matrix is required for proper shaping of the vulval lumen.

Overall, this is an important body of work as it establishes a new and powerful model with many advantages (live imaging, genetics, phenotypic readouts of function) to study poorly understood aECMs during epithelial lumen/tube formation. The work also has a number of important observations, such as how dynamic aECMs are (rapidly assembled and removed), their complex structures (fibrillar and granular components), specific cell type associations, their functions in both constricting and expanding lumens, and functional interactions with the chondroitin proteoglycan luminal matrix. Given the importance of aECMs in epithelial tube formation in vertebrates and invertebrates and the paucity of work examining aECMs, this represents an important advance for the field. Notably, in addition to the aECM, the authors also provide a complete characterization of vulval eversion, which is an overlooked aspect of vulval morphogenesis.

Essential revisions:

1) It would strengthen the manuscript if the authors can determine if a hemi-desmosome-like structures may link the aECM to the vulval cells-intracellular VAB-10a and receptors MUP-4 and MUA-3. We suggest to test if loss of VAB-10a, MUP-4 and MUA-3 result in vulval phenotypes that mirror the aECM loss.

2) Do loss of VAB-10a, MUP-4 and MUA-3 lead to changes in aECM localization?

---

## [Author Response]

Essential revisions:1) It would strengthen the manuscript if the authors can determine if a hemi-desmosome-like structures may link the aECM to the vulval cells-intracellular VAB-10a and receptors MUP-4 and MUA-3. We suggest to test if loss of VAB-10a, MUP-4 and MUA-3 result in vulval phenotypes that mirror the aECM loss.2) Do loss of VAB-10a, MUP-4 and MUA-3 lead to changes in aECM localization?

Unfortunately, we are not able to do any further experiments regarding VAB-10/MUP-4/MUA-3 at this time, and therefore we have toned down our statements regarding these factors. We submitted our manuscript April 14, your decision letter arrived June 26, and on June 30 first author Jennifer Cohen left to start her postdoc out of state. In fact, none of the co-authors of this manuscript remained in either the Sundaram or Frand labs as of July 1, and all PIs are still working remotely due to COVID-related health concerns.

We agree with the reviewers that understanding how the vulva aECM connects to cell membranes and the cytoskeleton will be very important for understanding how this aECM shapes cells and *vice versa*. However, we don't think that answering this question is central to this specific paper. Furthermore, it is a question that is likely to take significant long-term work to answer, given that the expression data shown in Figure 7 suggests that MUP-4 appears too late and is probably not a relevant linker for the pre-cuticle aECM factors we describe. Considering the spatial and temporal complexity of the observed aECM patterns, characterizing any number of transient attachments between vulva cells and interim matrices is going to require substantial additional work beyond the scope of this report.

We do plan to continue this line of investigation once we can return to the lab, and we would be pleased to publish our next advance in *eLife*, following the current policy.

In this revised manuscript, we now cite existing RNAi data for the relevant genes and use more cautious language in describing our conclusions:

"To address whether the known matrix-anchoring complexes could anchor aECMs in the vulva, we asked where VAB-10a and MUP-4 appear in vulval cells. […] Further experiments will be needed to test the roles of VAB-10, MUP-4 and MUA-3 more definitively, and to identify the mechanisms that link aECM to vulva apical membranes."